# Functional brain alterations following mild-to-moderate sensorineural hearing loss in children

**Axelle Calcus[1,2]\*, Outi Tuomainen[2], Ana Campos[2], Stuart Rosen[2], Lorna F Halliday[2,3]**

[1]Laboratoire des Systèmes Perceptifs, Département d'Etudes Cognitives, Ecole Normale Supérieure, PSL University, CNRS, Paris, France; [2]Department of Speech, Hearing and Phonetic Sciences, University College London, London, United Kingdom; [3]MRC Cognition and Brain Sciences Unit, University of Cambridge, Cambridge, United Kingdom

**Abstract** Auditory deprivation in the form of deafness during development leads to lasting changes in central auditory system function. However, less is known about the effects of mild-to-moderate sensorineural hearing loss (MMHL) during development. Here, we used a longitudinal design to examine late auditory evoked responses and mismatch responses to nonspeech and speech sounds for children with MMHL. At Time 1, younger children with MMHL (8–12 years; n = 23) showed age-appropriate mismatch negativities (MMNs) to sounds, but older children (12–16 years; n = 23) did not. Six years later, we re-tested a subset of the younger (now older) children with MMHL (n = 13). Children who had shown significant MMNs at Time 1 showed MMNs that were reduced and, for nonspeech, absent at Time 2. Our findings demonstrate that even a mild-to-moderate hearing loss during early-to-mid childhood can lead to changes in the neural processing of sounds in late childhood/adolescence.
DOI: https://doi.org/10.7554/eLife.46965.001

**\*For correspondence:**
axelle.calcus@ens.fr

**Competing interests:** The authors declare that no competing interests exist.

## Introduction

The structure and function of the adult auditory system is dependent upon stimulation received during maturation (*Kral and Sharma, 2012*). Animal studies have shown marked differences in the anatomy, physiology, and functionality of the central auditory pathway following cochlear ablation, pharmacological neonatal deafening, and congenital deafness (*Shepherd and Hardie, 2001*; *Berger et al., 2017*; see *Butler and Lomber, 2013*, for a review). Deafness-induced changes have also been identified in humans, namely, degeneration of spiral ganglion cells in the cochlea (*Nadol et al., 1989*), a reduction of volume of neurons in the cochlear nucleus (*Seldon and Clark, 1991*; *Chao et al., 2002*) and a functional reorganisation of cortical activity (*Neville et al., 1998*) following congenital profound (>95 dB HL) sensorineural hearing loss (SNHL). However, in contrast to the literature on deafness, much less is known about the effects of mild- (21–40 dB HL) to-moderate (41–70 dB HL; *British Society of Audiology, 2011*) SNHL (MMHL) on the developing auditory pathway. Nonetheless, there is evidence to suggest that even transient or mild levels of hearing loss during a critical period can lead to lasting effects in the central and peripheral auditory system (e.g. *Gravel et al., 2006*; *Rosen et al., 2012*; *Takesian et al., 2012*; *Liberman et al., 2015*; *Mowery et al., 2015*). Here, we examine the effects of MMHL on the development of auditory cortical functioning in children.

Cortical event-related potentials (ERPs) have been used extensively to investigate the functional integrity of the auditory pathway in both normally hearing (NH) and hearing-impaired individuals (for

a review, see *Alain and Tremblay, 2007*). Of particular interest are the late auditory evoked responses (LAERs), comprising the P1, N1, P2, and N2 components, and the mismatch responses (MMRs), comprising the mismatch negativity (MMN) and late discriminatory negativity (LDN). The LAERs are thought to mark the initial detection (P1, N1, and P2), classification and discrimination (P2 and N2) of an auditory stimulus (for a review, see *Eggermont and Ponton, 2002*). Typically, these components are elicited passively, in response to a stream of repeated standard sounds. In contrast, the MMRs are thought to reflect the discrimination of one or more different sounds, and are typically evoked by the detection of comparatively rare deviant auditory stimuli from a stream of repeated, comparatively frequent, standards (i.e. an oddball paradigm; e.g. *Patel and Azzam, 2005*). Both the LAERs and the MMRs show distinct postnatal maturational time courses (for reviews, see *Gomot et al., 2000*; *Ponton et al., 2000a*; *Bishop et al., 2010*). Therefore, these responses have been identified as useful candidates to assess the effects of SNHL on the development of the auditory pathway in children.

There is now an abundance of evidence to suggest that auditory cortical responses may be delayed and/or deviant in children with severe or profound SNHL (*Ponton et al., 1996a*; *Ponton et al., 1996b*; *Ponton et al., 1999*; *Ponton et al., 2000b*; *Ponton and Eggermont, 2001*; *Sharma et al., 2002*; *Sharma et al., 2005*; see *Näätänen et al., 2017*, for a review of the MMN). *Ponton et al. (1996b)* used a combined cross-sectional and longitudinal design to examine the development of the P1-N1-P2 complex in a group of six, 6–16 year-old children with early-onset profound SNHL who had been fitted with unilateral cochlear implants (CIs) following an average period of deafness of 4.5 years (range: 1.5 to 6 years). Note that this study was conducted prior to the introduction of newborn hearing screening programmes, and before changes in recommendations regarding the age and type (unilateral vs. bilateral) of cochlear implantation in children with profound SNHL (*Waltzman and Roland, 2005*; *Papsin and Gordon, 2007*). ERPs were elicited via electric pulse trains, and were compared to those of a group of NH control children, the ERPs of whom were evoked by acoustic pulse trains. For the youngest children (aged 5–7 years), the ERPs of the CI group were very similar to those of controls, being dominated by a large P1 around 140 ms after stimulus onset, followed by a prolonged negativity (N2). However, for the older children (8–16 years), marked morphological differences between groups emerged: Whereas older NH controls developed an N1 response, children with CIs generally did not. Instead, the ERPs of older children with CIs more closely resembled those of younger NH controls. *Ponton et al. (1999)* and *Ponton et al. (2000b)* examined a more extensive set of longitudinal ERP data recorded from CI users (N = 118) into their adolescence. P1 was found to emerge following implantation, albeit with a latency delay proportional to the period of deafness. Moreover, the MMN was robustly present in children with CIs who showed good spoken language perception (*Ponton et al., 2000b*). However, particularly striking was the absence of an N1 from almost all of the profoundly deaf children Ponton and colleagues had tested. *Ponton and Eggermont (2001)* argued that this provided evidence for a persistent immaturity in layer II of primary auditory cortex for children with congenital profound SNHL who were fitted (albeit often late) with CIs. More recently, *Singh et al. (2004)* measured the P1-N2 and MMN responses to /ba/-/da/ of 35, 7- to 17-year-old children with bilateral profound SNHL who had been fitted with CIs between 1 and 10 years. Following implantation, participants' hearing sensitivity was between 30–40 dB HL (i.e. equivalent to a mild SNHL) and, unlike in *Ponton et al. (2000b)*, stimuli were presented free-field at 75 dB SPL. As is typical for children with a hearing age of ≤10 years, ERP responses were dominated by a large positivity (P1) around 100 ms post-stimulus onset, followed by a large negativity (N2) peaking at around 240 ms. However, none of the children with CIs showed a significant N1, and only 10 (29%) showed an MMN.

Like children with severe-to-profound SNHL, those with MMHL also experience reduced and degraded auditory input. However, the few studies that have assessed LAERs and MMRs for children with MMHL have yielded inconsistent results. *Rance et al. (2002)* measured unaided P1-N1-P2 responses to speech (synthesised /dæd/) and nonspeech (440 Hz pure tone) sounds in 18, 3-to-9-year-old children with SNHL, of which five had MMHL. All the children with MMHL showed ERP responses to both stimulus types, the amplitude and latency of which did not differ from those of NH controls. *Martinez et al. (2013)* recorded the acoustic change complex (ACC) to changes in vowel height and place contrasts in five 2–6 year-old children with MMHL. Robust and age-appropriate P1-N2 responses were observed in all but one of the children with MMHL whilst they were wearing their hearing aids. Of the two children who were also tested with their hearing aids turned off,

one showed an ACC response that was markedly reduced, while the other failed to show a response at all (*Martinez et al., 2013*). To our knowledge, only one study has examined the effects of congenital MMHL on both the LAER and the MMN in children (*Koravand et al., 2013*). The unaided ERPs of 12, 9–12 year-old children with MMHL were compared to those of 16 NH, age-matched controls. Stimuli were speech (synthesized /ba/ and /da/), simple nonspeech (1- and 2-kHz pure tones), and speech-like complex nonspeech sounds (acoustic transformations of /ba/ and /da/), presented between 80–100 dB HL for children with MMHL. Whereas all children showed P1 and N2 components, N1 and P2 responses were absent in almost half of the children with MMHL, compared to only a quarter of NH controls. The amplitude and latency of P1 and the MMN did not differ between children with MMHL and NH controls. However, across all stimuli, the N2 was significantly smaller (but not later) in children with MMHL. The authors concluded that reduced N2 amplitude may be a neurophysiological marker of central auditory processing deficits in children, including those with MMHL.

It is not immediately apparent why some studies have observed age-appropriate LAERs in children with MMHL (*Rance et al., 2002*; *Martinez et al., 2013*), while others have not (*Koravand et al., 2013*). However, it is notable that where differences have been reported, these have been confined to older (9–12 year-old; *Koravand et al., 2013*) rather than younger (2–9 year-old; *Rance et al., 2002*; *Martinez et al., 2013*) children. Interestingly, delayed and diminished N1, N2, and MMN responses to speech sounds have also been observed in adults with MMHL, particularly at lower stimulus intensities (*Oates et al., 2002*). One possible interpretation of these findings therefore is that, as for severe-to-profound SNHL (*Ponton et al., 1996a*; *Ponton et al., 1996b*; *Ponton et al., 1999*; *Ponton et al., 2000b*; *Ponton and Eggermont, 2001*), the effects of MMHL on the neural processing of sounds may only become apparent as children grow older.

The present study used a cross-sectional followed by longitudinal design to investigate the effects of MMHL on the development of LAERs and MMRs across mid-childhood and adolescence. At Time 1, we measured the LAERs and MMRs evoked by simple nonspeech, complex speech-like nonspeech, and speech stimuli for 46, 8–16 year-old children with MMHL (MM group) and 44 NH age-matched controls (NH group). Stimuli were presented at a constant intensity of 70 dB SPL, and children with MMHL were tested without their hearing aids in order to avoid stimulus alterations being introduced by the different hearing aid processors (*Billings et al., 2007*). In order to evaluate developmental effects, children were divided into younger (Y; 8–11 years) and older (O; 12–16 years) age bands (see *Table 1*), consistent with reports of step-function changes in the LAER in NH children between those ages (*Bishop et al., 2007*). There were no differences in audiometric thresholds between the Y and O children with MMHL (see *Figure 1*). However, whereas younger children with MMHL exhibited age-appropriate MMNs to all three stimulus conditions, the MMN was absent in older children with MMHL. At Time 2 (+ 6 years), we re-tested a subgroup (n = 13) of children from the MM-Y group from Time 1. Owing to the time-lag between experiments, these children were now aged 13–17 years (i.e., they would now qualify for the MM-O group from Time 1). Despite showing age-appropriate MMNs at the group level when they were younger, these children showed no MMN to nonspeech or speech-like sounds six years later. Our findings therefore suggest that MMHL in early-to-mid childhood leads to functional changes in auditory cortex that are predominantly reflected in the MMN, and that only emerge during adolescence.

## Results

### Cross-sectional study (Time 1)

#### Children with MMHL show reduced presence of LAER components to sounds

To assess auditory cortical functioning in children with MMHL, we recorded the EEG from all children during a passive oddball paradigm for three stimulus conditions (simple nonspeech (*nonspeech*), complex speech-like nonspeech (*speech-like*), and speech sounds (*speech*)), and three stimulus types (*standards, large* (easy-to-detect) *deviants*, and *small* (difficult-to-detect) *deviants*; see *Figure 2* and Methods). *Figure 3* shows the grand-average waveforms evoked at electrode Cz for each condition (nonspeech vs. speech-like vs. speech), stimulus type (standards vs. large deviants vs. small deviants), group (NH vs. MM) and age band (Y vs. O). As is typical for younger NH children in this age range

**Table 1.** Mean (SD) participant characteristics for the NH and MM groups and Y and O age bands (cross-sectional study)

| | Younger (Y) | | Older (O) | |
|---|---|---|---|---|
| Variable | NH-Y (n = 26) | MM-Y (n = 23) | NH-O (n = 18) | MM-O (n = 23) |
| **Demographics** | | | | |
| Age (years)[a] | 10.1 (1.0) | 9.6 (1.3) | 13.6 (1.2) | 13.2 (1.0) |
| Nonverbal IQ (T score)[b] | 60.4 (10.0) | 58.3 (9.2) | 61.0 (5.8) | 53.0 (7.5) |
| Maternal education (years) | 20.4 (2.3) | 19.3 (2.7) | 20.6 (3.6) | 19.3 (2.7) |
| **Audiometry** | | | | |
| BEPTA threshold (dB HL)[b] | 8.07 (4.4) | 44.1 (10.8) | 6.9 (4.3) | 42.6 (13.3) |
| WEPTA threshold (dB HL)[b] | 11.2 (5.4) | 50.7 (12.6) | 9.8 (4.4) | 46.5 (12.9) |
| MePTA threshold (dB HL)[b] | 9.2 (5.0) | 49.4 (10.1) | 7.8 (4.3) | 46.9 (13.5) |
| **Hearing loss history** | | | | |
| Age of detection (months) | | 49.9 (25.1) | | 59.0 (44.1) |
| Hearing aids (n; %) | | 23; 100% | | 18; 78.2% |
| Age of aiding (months) | | 59.4 (26.8) | | 69.6 (56.3) |

*Note.* NH = age-matched normally hearing control group; MM = mild-to-moderate SNHL group; Y = younger; O = older; Age = mean age (years) at session 1 and session 2; Nonverbal IQ = T score on Block Design subtest of the Wechsler Abbreviated Scale of Intelligence (**Wechsler, 1999**); Maternal education = age (years) at which mother left full-time education; BEPTA = better-ear pure-tone average across octave frequencies 0.25–4 kHz; WEPTA = worse-ear PTA across octave frequencies 0.25–4 kHz; MePTA = Mean PTA across octave frequencies 0.25–8 kHz for left and right ears; Age of detection = age (months) at which SNHL was detected; Age of aiding = age (months) at which hearing aids were first fitted. Significant main effects of age band, group, and their interaction are denoted by [a, b] and [c] respectively (see Cross-sectional study, Participants, for details).

DOI: https://doi.org/10.7554/eLife.46965.003

(**Sussman et al., 2008**), the LAER to standards for the NH-Y subgroup was dominated by a large positivity, occurring around 100 ms post-stimulus-onset (P1), followed by a large, prolonged negativity from around 150–600 ms (N2). For the NH-O subgroup, the LAER was characterised by a P1 (around 100 ms), N1 (~130 ms), P2 (~180 ms), and N2 (~250–300 ms), which were particularly marked for the nonspeech condition.

To determine whether children with MMHL were less likely to show LAER components to standards than NH children, presence or absence of components P1, N1, P2, and N2 at electrode Cz was identified for each individual and condition by three independent judges (see Methods). *Figure 4* shows the percentage of children for each group, age band, and condition showing a P1, N1, P2, or N2 response (see also *Figure 4—figure supplement 1*). A series of mixed-effects logistic regression models was applied to determine whether group, age band, condition, component, or their interactions predicted presence/absence of components (see Statistical analyses, scripts available on https://github.com/acalcus/MMHL.git, and *Appendix 1—table 1*). Children with MMHL were less likely to show present components than NH children [odds ratio (OR): 0.24, p<1e$^{-15}$]. In addition, regardless of group, children in the O age band were more likely to show a P2 than those in the Y age band [OR: 1.71, p=0.038].

## Older children with MMHL show an LDN but not MMN to deviant sounds

To assess whether children with MMHL showed evidence for neural discrimination of auditory deviants, MMNs and LDNs were calculated for each group, age band, condition, and deviant type (large

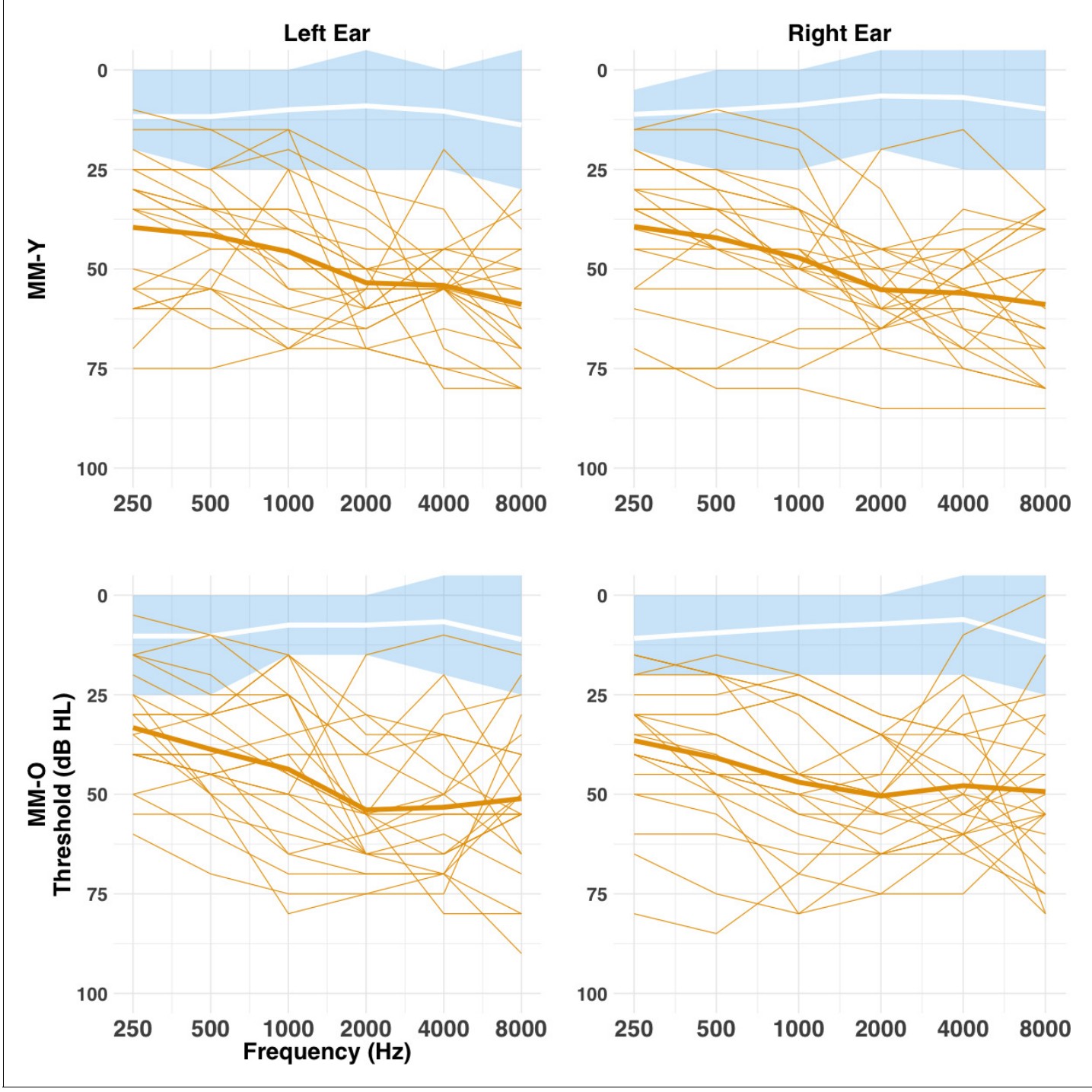

**Figure 1.** Pure-tone air-conduction audiometric thresholds (cross-sectional study). Audiometric thresholds are shown across octave frequencies from 0.25 to 8 kHz for the MM (orange) and NH (blue and white) groups, Y (top row) and O (bottom row) age bands, and left (left column) and right (right column) ears. Individual thresholds for the MM group are shown as normal lines, and the group mean marked by a bold line. Mean thresholds for the NH group are marked by a white bold line, with the shaded blue area representing the range for the NH group. There was no difference between the MM-Y or MM-O subgroups in audiometric thresholds (all p>0.05; see text).

DOI: https://doi.org/10.7554/eLife.46965.002

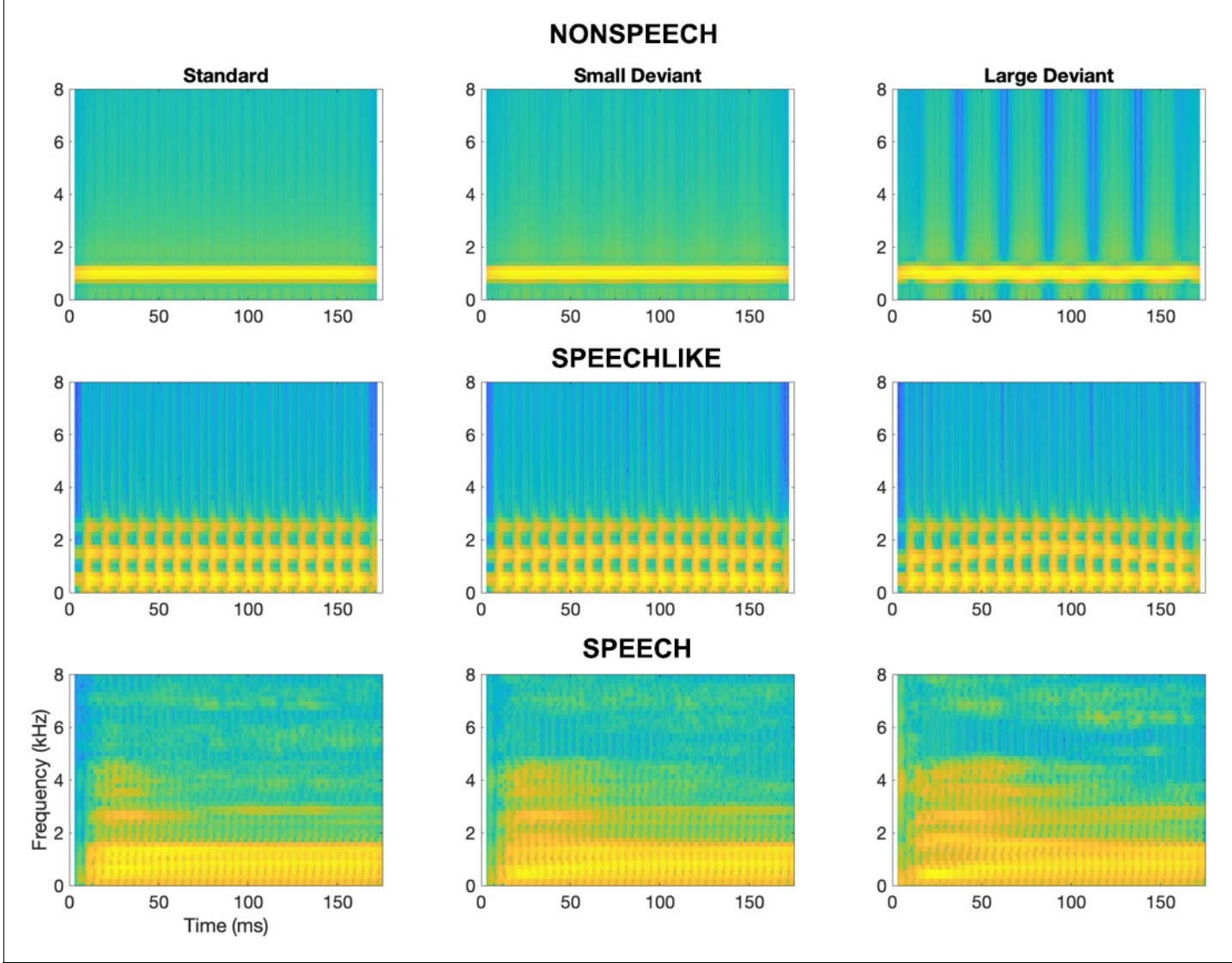

**Figure 2.** Spectrograms depicting stimuli for the nonspeech, speech-like, and speech conditions. For each condition, the left panel depicts the standard stimulus, the middle panel depicts the small deviant, and the right panel depicts the large deviant.

DOI: https://doi.org/10.7554/eLife.46965.004

and small). To do so, point-to-point comparisons (unilateral *t*-tests) of the differential (standard minus deviant) wave amplitudes were performed to determine the latency period over which the waveforms were significantly different from zero (see Methods). Because adjacent points in the waveform are highly correlated, potentially leading to spurious significant values over short intervals, a response was considered present when p<0.01 (one-tailed) for >20 ms at adjacent time-points (*Kraus et al., 1993*; *McGee et al., 1997*). The 100–400 ms post-stimulus-onset window was selected for the MMN, and the 400–600 ms window for the LDN (*Bishop et al., 2007*). Calculations were performed at the group level, because the MMN and LDN are known not to be reliable at the level of the individual (*Picton et al., 2000*; *Dalebout and Fox, 2001*; *Bishop and Hardiman, 2019*).

*Figure 3* also shows the grand averages for the small and large deviants for each group, age band, and condition, along with the periods of the post-stimulus window that showed a significant MMN or LDN (see also *Appendix 1—table 2*). At the group level, both the Y and O NH subgroups showed a significant MMN to large deviants in all three conditions, and a significant LDN to large deviants in the speech-like and speech conditions. Similarly, the MM-Y subgroup showed a significant MMN to large deviants in all conditions, a significant but brief MMN to small deviants in the

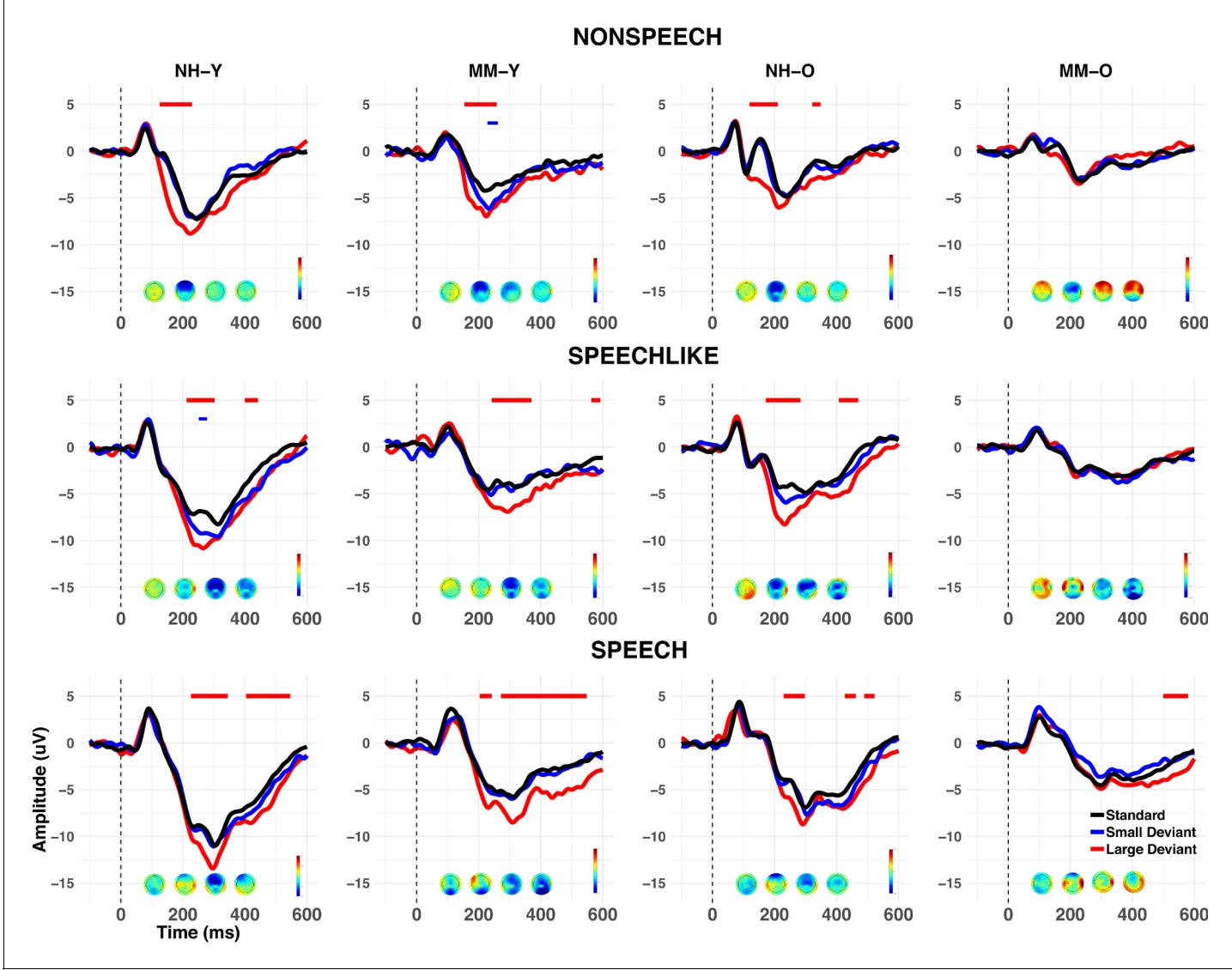

**Figure 3.** Grand average waveforms at Cz (cross-sectional study). Grand averages are shown for each group (NH vs. MM), age band (Y vs. O), and condition (nonspeech, speech-like, speech). Responses to standards are shown in black, and to large and small deviants in red and blue respectively. Voltage maps show the mean MMN activity during the 100–400 ms post-stimulus time window. Negative values of the MMN are shown in blue, and positive values in red. Periods of the 100–400 ms (MMN) and 400–600 ms (LDN) post-stimulus epoch where a significant (p<0.01 for>20 ms consecutively) MMR was observed in the grand average are shown by horizontal lines above the waveforms, as a function of deviant type. Note that whilst both the Y and O NH subgroups (NH-Y and NH-O), and the MM-Y subgroup obtained significant MMNs for large deviants across all conditions, the MM-O subgroup failed to show a significant MMN to any stimulus type in any condition (see text for details).

DOI: https://doi.org/10.7554/eLife.46965.005

nonspeech condition, and a significant LDN to large deviants in the speech-like and speech conditions. However, contrary to their Y or NH peers, the MM-O subgroup failed to show a significant MMN to either large or small deviants in any of the three conditions, or an LDN to large or small deviants in the nonspeech or speech-like conditions. Rather, the MM-O subgroup only showed a significant LDN to large deviants in the speech condition. Because small deviants rarely elicited an MMR for any of the subgroups or conditions, they were not included in the subsequent analyses.

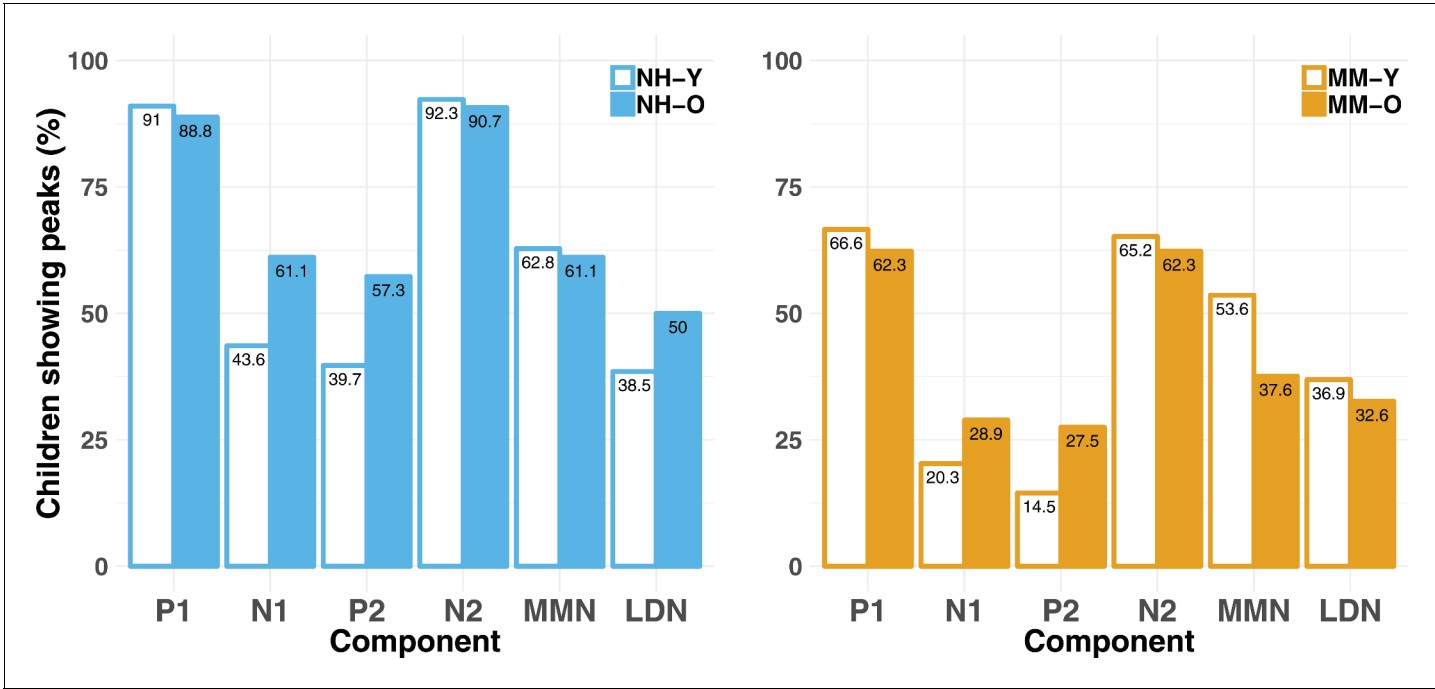

**Figure 4.** Percentages of children showing present LAER components (P1, N1, P2, and N2) and MMN/LDN responses (cross-sectional study). Percentages are shown for each age band (Y and O) and for each group (NH and MM), for each component (P1, N1, P2, N2) and response (MMN and LDN). Percentages are averaged across the three conditions (nonspeech, speech-like, speech). Regardless of age band, children in the MM subgroups were less likely to show present LAER components and MMNs (but not LDNs) than their NH peers.

DOI: https://doi.org/10.7554/eLife.46965.006

The following figure supplements are available for figure 4:

**Figure supplement 1.** Probability of presence of the LAER components (P1, N1, P2, N2) for each group (MM vs. NH) as a function of age (cross-sectional study).

DOI: https://doi.org/10.7554/eLife.46965.007

**Figure supplement 2.** Probability of presence of the MMN and LDN for each group (MM vs. NH) as a function of age (cross-sectional study).

DOI: https://doi.org/10.7554/eLife.46965.008

## Older children with MMHL show smaller MMNs but not LDNs to deviant sounds

The absence of an MMN for the MM-O subgroup may have several different explanations. For instance, the MM-O subgroup may have (a) been individually less likely to demonstrate MMNs to deviant sounds, (b) shown MMNs that were later or more variable in latency, or (c) shown MMNs that were reduced in amplitude, relative to their NH peers. To determine whether children in the MM-O subgroup were less likely to show an MMN to large deviants than NH children, presence or absence of a significant MMN or LDN was identified for each participant, for each condition (see Methods). Because MMRs are less reliable at the individual level, an adjusted value of p<0.05 (one-tailed) for >20 ms at adjacent time-points was used. *Figure 4* also shows the percentage of children for each group, age band, and condition that showed a significant MMN or LDN to large deviants. Logistic regressions were used to ascertain whether group, age band, condition, or their interactions predicted presence of the MMN or LDN (see *Appendix 1—table 1* and *Figure 4—figure supplement 2*). Regardless of age band, children with MMHL were significantly less likely to show an MMN than NH children [OR = 0.51, p=0.007]. There were no significant main effects or interactions for the LDN.

To determine whether the MM-O subgroup showed MMNs that were delayed in onset latency, a linear mixed-effects model was conducted to determine whether group, age band, condition, or their interactions predicted MMN latencies to large deviants. Only the main effect of condition was significant ($\chi(2)$=39.69, p<0.001), with MMN onset earlier in the nonspeech condition than both the

**Table 2.** Mean (SD) participant characteristics for the four subgroups (MM-Y, MM-YO, NH-O and MM-O; longitudinal study)

| | Younger (Y) | | Older (O) | |
| --- | --- | --- | --- | --- |
| Variable | MM-Y (n = 13) | MM-YO (n = 13) | NH-O (n = 18) | MM-O (n = 23) |
| **Demographics** | | | | |
| Age (years) | 9.5 (1.3)[a] | 14.8 (1.4)[b] | 13.6 (1.2)[c] | 13.2 (1.0)[c] |
| Nonverbal IQ (T score) | 59.6 (10.2) | - | 61.0 (5.8) | 53.0 (7.5) |
| Maternal education (years) | 19.2 (2.8) | 19.2 (2.8) | 20.6 (3.6) | 19.3 (2.7) |
| **Audiometry** | | | | |
| BEPTA threshold (dB HL) | 40.1 (9.0)[a] | 35.7 (9.0)[a] | 6.9 (4.3)[b] | 42.6 (13.3)[a] |
| WEPTA threshold (dB HL) | 49.8 (14.1)[a] | 49.8 (18.2)[a] | 9.8 (4.4)[b] | 46.5 (13.0)[a] |
| MePTA threshold (dB HL)[b] | 46.7 (9.3)[a] | 44.9 (11.4)[a] | 8.8 (4.5)[b] | 45.5 (12.4)[a] |
| **Hearing loss history** | | | | |
| Age of detection (months) | 50.6 (23.6) | - | - | 59.0 (44.1) |
| Hearing aids (n; %) | 13; 100% | - | - | 18; 78.2% |
| Age of aiding (months) | 63.6 (23.2) | - | - | 69.6 (56.3) |

*Note.* MM = mild-to-moderate SNHL group; Y = younger; MM-YO = children from the MM-Y subgroup from the cross-sectional study (Time 1) who were followed-up as part of the longitudinal study (Time 2). NH = normally hearing control group; O = older; Age = mean of session 1 and session 2 (years); Nonverbal IQ = T score on Block Design subtest of the Wechsler Abbreviated Scale of Intelligence (**Wechsler, 1999**); Maternal education = age (years) at which mother left full-time education; BEPTA = better-ear pure-tone average across octave frequencies 0.25–4 kHz; WEPTA = worse-ear pure-tone average across octave frequencies 0.25–4 kHz; MePTA = Mean PTA across octave frequencies 0.25–8 kHz for left and right ears; Age of detection: age (months) at which SNHL was detected (MM subgroups only); Age of aiding = age (months) at which hearing aids were first fitted. Subgroups that differed significantly from one another on a given variable are denoted by [a], [b] and [c] (see Longitudinal study, Participants).

DOI: https://doi.org/10.7554/eLife.46965.016

speech-like and speech conditions (both p<0.001), which did not differ from each other (p=0.863). To determine whether the MM-O subgroup showed greater variability in MMN onset latency, Levene's tests for equality of variance were used. There were no significant differences in variance for group, age band [both $Fs$ <1, both $ps$ >0.10], or their two-way interaction [$F$ = 1.45, p=0.229].

Finally, to determine whether children in the MM-O subgroup had MMRs that were reduced in amplitude relative to their NH peers, MMN and LDN amplitudes were calculated from the differential waveforms for each participant and condition, for the large deviants only (see Methods). Linear mixed-effects models were then conducted to determine whether group, age band, condition, or their interactions predicted MMN or LDN amplitudes to large deviants (see *Appendix 1—table 3*). For the MMN, the group × age band interaction contributed to the final model, just missing significance (p=0.069). Post-hoc $t$-tests indicated that whereas the NH-Y and MM-Y subgroups showed a small but non-significant difference in their MMN amplitudes [$t$(135.45)=−1.82, p=0.071, CI (−2.27, 0.09)], the MM-O subgroup had significantly smaller MMNs than the NH-O subgroup [$t$(120.56)=−5.15, p<0.001, CI (−3.84,−1.71)] (see *Figure 5*). Difference wave amplitudes over the LDN time window did not differ between groups or age bands.

## LAERs and MMNs, but not LDNs, are smaller or later for children with MMHL, when present

Because children with MMHL were less likely to show LAERs and MMNs than their NH peers, we then asked whether these responses were normal for those children who did show them. To do so, we calculated the latencies and amplitudes of those components/responses that had been identified as present for each individual (see Methods and *Figure 5—figure supplement 1*). Linear mixed-models were then run to assess whether group, age band, condition, or their interactions, predicted the latencies and/or amplitudes of each component/response (see *Appendix 1—table 4*). Note that these analyses represent the best-case-scenario for children with MMHL, in that only a subset of children were included in any given analysis (see *Figure 4*). Regardless of age, where present, P1 and P2 were slightly later (both by 11 ms; p=0.012, and $p$ = .051, respectively) but not smaller (p=0.849 and p=0.468, respectively) in children with MMHL compared to NH controls. In contrast, N2 was

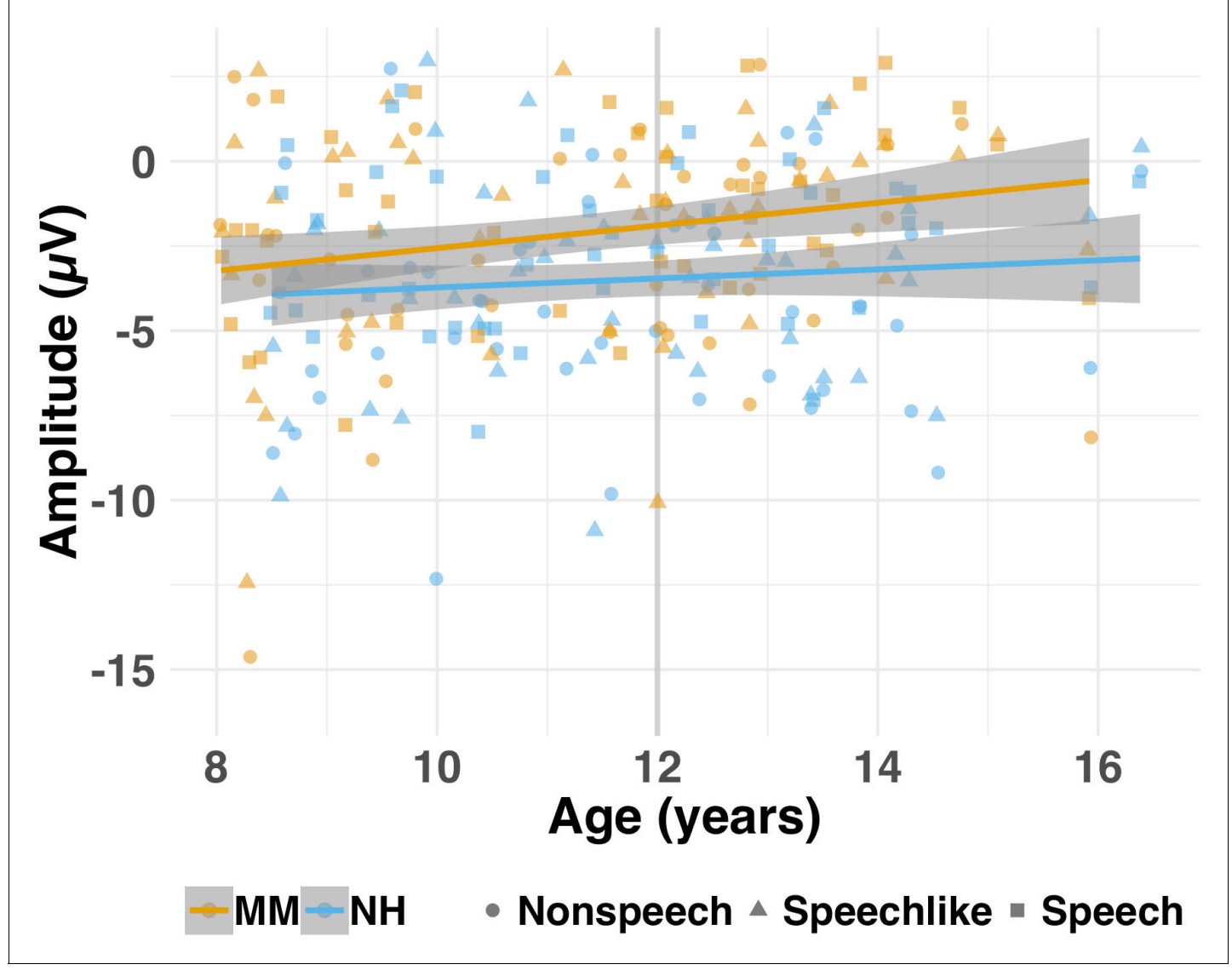

**Figure 5.** Amplitude (μV) of the MMN for each group (MM vs. NH) as a function of age (cross-sectional study). Individual (shapes) and group (lines) data are shown for each condition (nonspeech = circles; speech-like = triangles, speech = squares) for the MM (orange) and NH (blue) groups. Shaded lines represent the 95% confidence interval. For illustrative purposes, age is represented as a continuous variable, but was treated as a categorical variable (Y vs. O) in the analyses (see text). The grey vertical lines on each subplot represent the median age split into the Y (left of the line) and O (right of the line) age bands. Children in the MM-O subgroup obtained MMNs that were smaller in amplitude than those of their NH peers (NH-O subgroup).

DOI: https://doi.org/10.7554/eLife.46965.009

The following figure supplement is available for figure 5:

**Figure supplement 1.** Amplitude (μV) and latency (ms) of the LAER components (P1, N1, P2, and N2), where present, for each group (MM vs. NH) as a function of age (cross-sectional study).

DOI: https://doi.org/10.7554/eLife.46965.010

significantly smaller (by 2.67 μV; p<0.001) but not later in children with MMHL ($p$ = .482), an effect that was greater in the speech-like (p<0.001) than both the nonspeech and speech conditions (p=0.014 and p=0.002, respectively). Neither the amplitude or latency of the N1 differed between groups, where present (p>0.10). However, note that only around ¼ of children with MMHL showed an N1 or P2 response.

For those individuals who showed a significant MMN to large deviants, the group × age band interaction for MMN amplitude again just missed significance (p=0.055). Again, this was driven by the absence of a significant group difference for the Y subgroups [$t$(69.57)=−0.14, p=0.891, CI

(−1.29, 1.13)], in the presence of significantly smaller MMNs for children in the MM-O subgroup relative to their NH-O peers [$t(48.46)=−3.20$, p=0.002, CI (−3.01,–0.69)]. For the LDN, there were no differences between groups or age bands in the amplitude of the response where present (p>0.10). However, again, the LDN was only present in around ⅓ of children with MMHL.

## Worse MMHL is associated with reduced presence and amplitude of LAERs and MMNs

Because stimuli in the current study were presented at a fixed presentation level, it is possible that group differences were due in part to the reduced sensation level experienced by the MM group. In order to determine whether severity of MMHL predicted presence/absence of components, a series of mixed-effects logistic regression models was applied to determine whether better-ear pure-tone average (BEPTA) audiometric thresholds, age band, condition, or component predicted the presence of LAER or MMR components in children with MMHL (see *Figures 6* and *7*, and *Appendix 1—table 5*). For the LAER, worse BEPTA thresholds were associated with reduced likelihood of presence of components in all conditions, although the effect was greater for the P1 and N2 (both $ps <1e^{−9}$) than the N1 and P2 components (both $ps <10e^{−6}$). Worse BEPTA thresholds were also associated with reduced likelihood of presence of an MMN (p=0.020), but this was consistent across age bands. For the LDN, there was a significant age band $\times$BEPTA threshold interaction (p=0.039). However, post-hoc logistic regressions conducted separately for each age band failed to reveal a significant effect of BEPTA on either the Y (p=0.245) or O age bands (p=0.089).

To determine whether severity of MMHL predicted latency/amplitude of components, where present, a series of mixed-effects linear regressions was conducted on present components/ responses with BEPTA thresholds as a continuous variable (see *Figure 8*, and *Appendix 1—table 6*). Worse BEPTA thresholds were associated with smaller (but not later) P1 amplitudes for the speech-like condition only (p=0.037), and later (but not smaller) N1s across conditions. Regardless of age band, where present, MMN amplitudes decreased with worsening MMHL (p=0.003; see *Figure 9*). There was no significant effect of BEPTA on the amplitude of the LDN, where present (p=0.326).

## Longitudinal follow-up (Time 2)

The cross-sectional study showed that, regardless of age, children with MMHL were less likely to show LAER components than their NH peers. Moreover, whereas younger children with MMHL showed a significant MMN to large deviants in all conditions and an LDN in the speech-like and speech conditions, older children with MMHL did not show an MMN in any condition, and only showed an LDN to speech. This *disappearing MMN* appeared to be driven by a greater reduction in MMN amplitude for the older children with MMHL relative to their younger peers.

The MMN findings for the MM-O subgroup imply two alternative hypotheses. First, it is possible that the MM-O subgroup at Time 1 was disadvantaged relative to the MM-Y subgroup, perhaps reflecting a difference in the quality of, or access to, intervention between children born at an earlier time and those born later. Indeed, while the MM-Y and MM-O subgroups did not differ in severity of SNHL, age of detection, or age of aiding (see *Figure 1* and *Table 1*), children in the MM-O subgroup were less likely to have or wear hearing aids than those in the MM-Y subgroup (see Methods and *Table 1*). Moreover, a higher proportion of those in the younger age band were born after the introduction of universal newborn hearing screening in the UK. Second, it is possible that MMHL in children leads to progressive degeneration in the structure and/or function of the auditory pathway.

To test these hypotheses, a subset of children from the initial MM-Y subgroup at Time 1 was retested 6 years later (Time 2). Owing to the time-lag between experiments, these children were all aged 14–17 years at Time 2 (i.e. they would now qualify for the MM-O subgroup of the cross-sectional study). Consequently, at Time 2, these children were referred to as the *MM-YO* subgroup. Crucially, the MM-YO subgroup were part of a larger group (MM-Y group from the cross-sectional study) that showed significant MMNs to large deviants at Time 1. Moreover, children in the MM-YO subgroup all wore hearing aids, and their audiometric thresholds did not deteriorate between test and re-test (see *Table 2*, Methods and *Figure 11—figure supplement 1*). If the differences between the MM-Y and MM-O subgroups observed in the cross-sectional study were due to historical changes in quality of care, then children in the MM-YO subgroup should continue to show an MMN to large deviants at Time 2, whatever the condition. Conversely, if the differences were due to

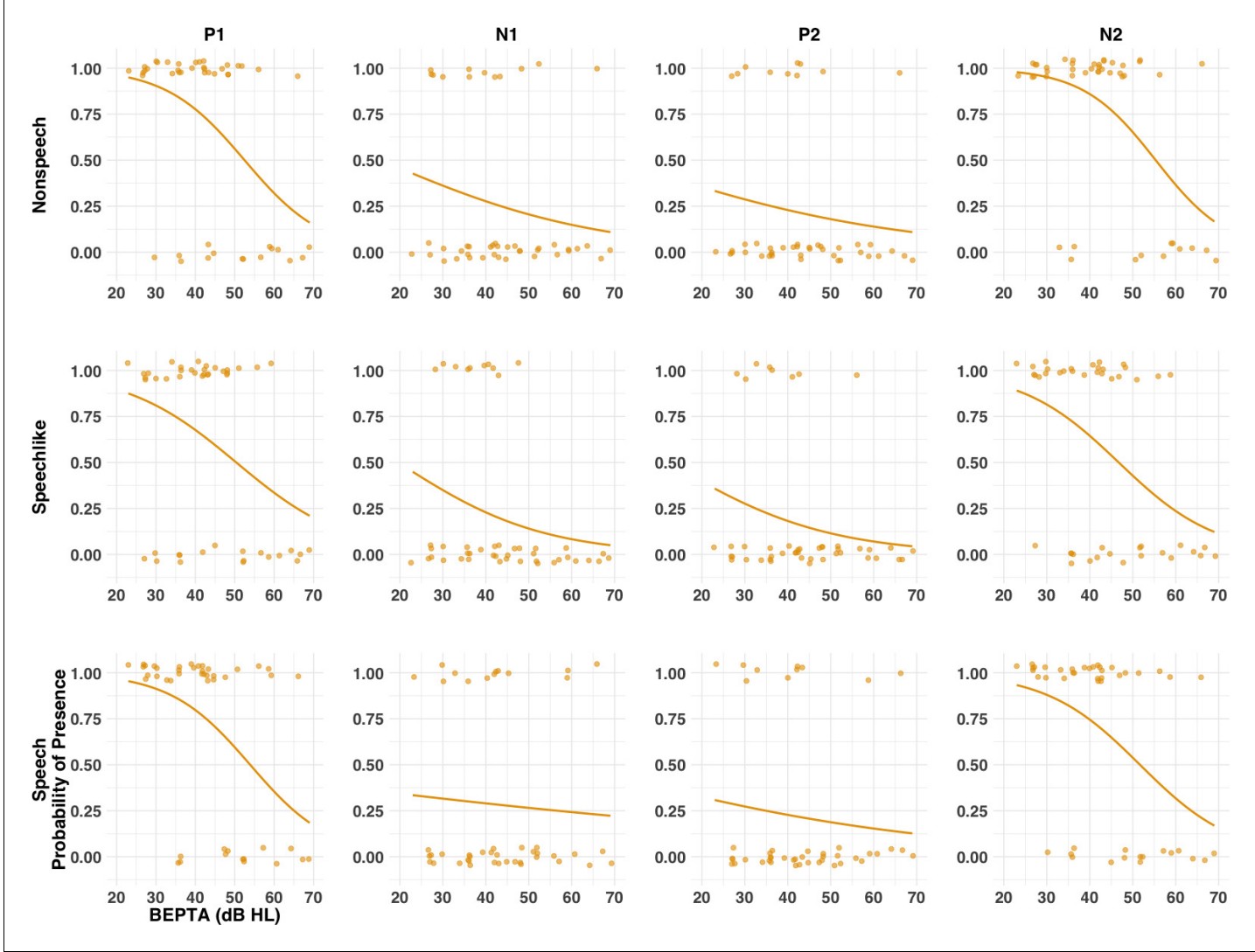

**Figure 6.** Probability of presence of LAER components (P1, N1, P2, and N2) for the MM group as a function of severity of hearing loss (cross-sectional study). Individual (circles) and group (lines) data are shown each condition (nonspeech, speech-like, speech) and component. At the individual level, components were either present (probability = 1) or absent (probability = 0). Solid lines result from a logistic regression fitting a general linear model with BEPTA as a predictor and presence of the component as the outcome variable. Worse BEPTA thresholds were associated with reduced likelihood of presence of components in all conditions, although this effect was more marked for the P1 and N2 components than for N1 and P2.

DOI: https://doi.org/10.7554/eLife.46965.011

developmental changes as a result of childhood MMHL, then children in the MM-YO subgroup should no longer show an MMN at Time 2.

## Children with MMHL show decreasing presence of P1 and N2 components with age

To assess changes in the development of LAER components with age, we compared the presence/absence of these components for the MM-YO subgroup at Time 2 to their younger selves (i.e. a subset of the MM-Y subgroup from Time 1). *Figure 10* shows the grand-average waveforms at Cz for each condition and stimulus type for the MM-YO and MM-Y subgroups, along with their historical peers (MM-O subgroup from Time 1) and their age-matched controls (NH-O subgroup from Time 1). In order to examine the effects of MMHL on the development of components of the LAER, the proportions of P1, N1, P2, and N2 components present for the MM-YO group were compared to when they were younger (see *Figure 11*). The proportion of children in the MM-YO group showing P1 and

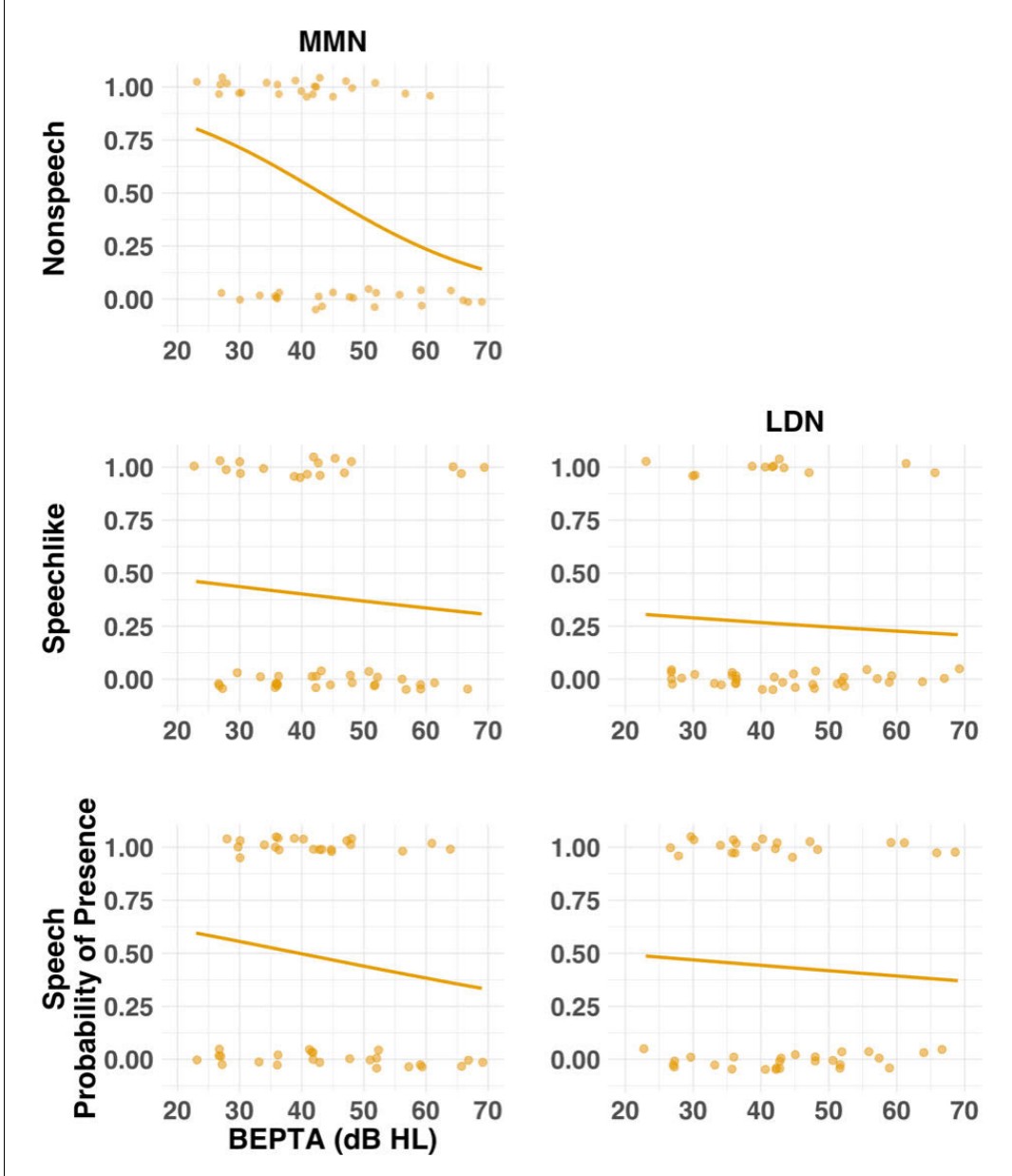

**Figure 7.** Probability of presence of the MMN and LDN for the MM group as a function of severity of hearing loss (cross-sectional study). Individual (circles) and group (lines) data are shown for each condition (nonspeech, speech-like, speech) and response where present for the NH group. Solid lines result from a logistic regression fitting a general linear model with BEPTA as a predictor and presence of the response as the outcome variable. Worse BEPTA thresholds were associated with reduced likelihood of presence of the MMN and LDN in all conditions.
DOI: https://doi.org/10.7554/eLife.46965.012

N2 components decreased from Time 1 to Time 2 [OR = 0.52, $\chi^2(1)$=5.37, p=0.021; and OR = 0.43, $\chi^2(1)$=5.49, p=0.019, respectively]. Conversely, the proportion of the MM-YO group showing a P2 increased between time-points [OR = 4.32, $\chi^2(1)$=6.35, p=0.011].

## Children with MMHL show decreasing presence of the MMN but not LDN with age

To assess longitudinal changes in the MMRs with age, we identified the periods of the 100–600 ms post-stimulus epoch where a significant MMN or LDN was observed for each subgroup, condition, and deviant type (see *Figure 10* and *Appendix 1—table 7*). In the nonspeech condition, although a

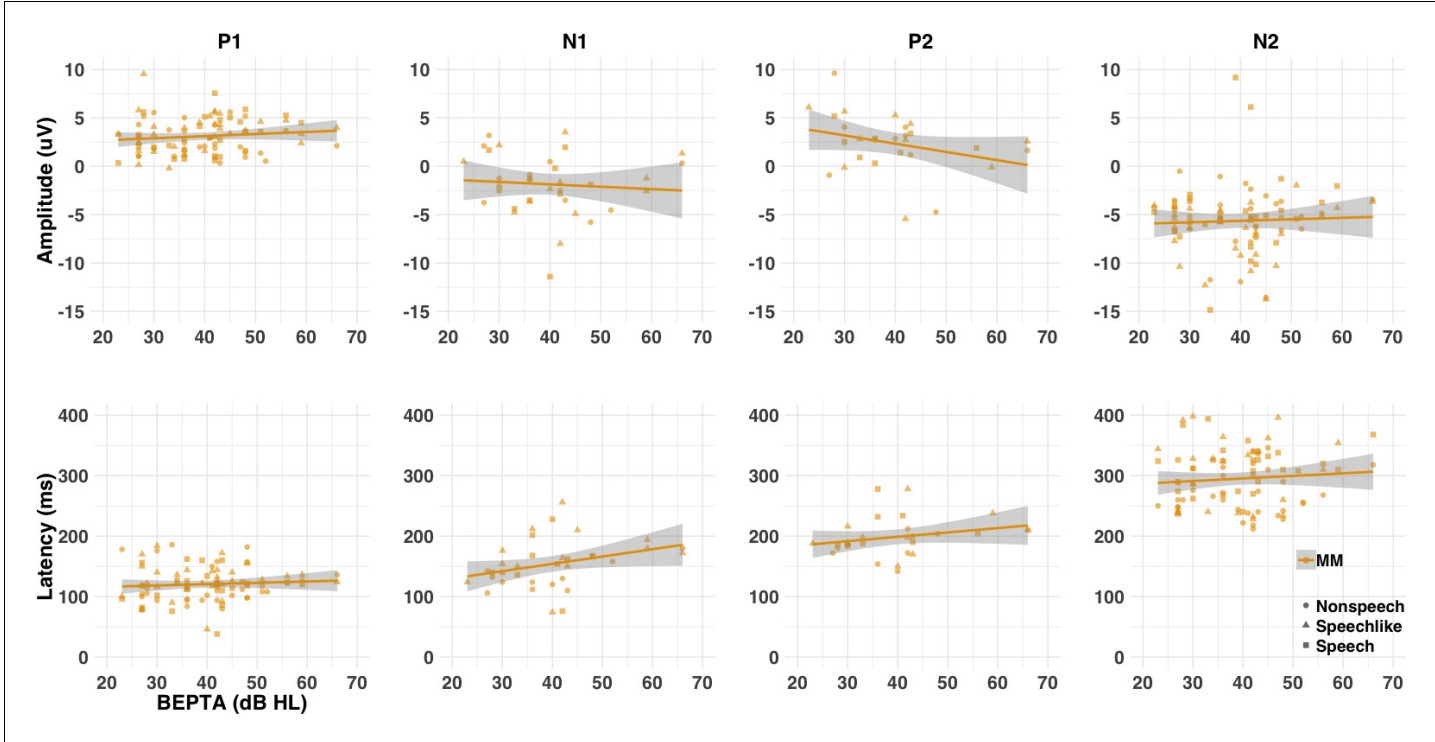

**Figure 8.** Amplitude (µV) and latency (ms) of LAER components (P1, N1, P2, and N2) for the MM group as a function of severity of hearing loss (cross-sectional study). Individual (shapes) and group (lines) data are shown for each condition (nonspeech = circles; speech-like = triangles; speech = squares) and component. Shaded lines represent the 95% confidence interval. Worse BEPTA thresholds were associated with smaller P1 amplitudes in the speech-like condition, and later N1s across conditions, where present.

DOI: https://doi.org/10.7554/eLife.46965.013

significant MMN was observed for the MM-Y group at Time 1, this was no longer the case at Time 2 (MM-YO group). In the speech-like condition, no significant MMN was observed for either the MM-Y group or their MM-YO counterpart. Note that this is unlike the MM-Y group from the cross-sectional study, where a significant MMN was observed for the speech-like condition (see *Figure 3*). In the speech condition, significant MMNs and LDNs remained for the MM-YO group.

Developmental changes to the MMN following MMHL were investigated in four ways (see *Appendix 1—table 8*), yielding four main results. First, children in the MM-YO subgroup were less likely to show an MMN than their younger selves (MM-Y) [OR = 0.38, $\chi^2(1)$=4.24, p=0.039]. Second, children in the MM-YO subgroup were less likely to show an MMN than children in the NH-O subgroup [OR = 0.17, p<0.001] but were not significantly different from those in the MM-O subgroup [OR = 1.83, p=0.148]. Third, the MMN was significantly smaller (on average by 1.49 µV) for the MM-YO subgroup than for their younger selves (p=0.028). Finally, while MMN amplitude for the MM-YO subgroup did not differ significantly from that of the initial MM-O subgroup (p=0.957), it was significantly smaller than that observed for the NH-O subgroup from Time 1 (p<0.001).

## Discussion

This study investigated the effects of permanent, MMHL on the development of both the LAER and MMRs across mid- to late-childhood. Children with MMHL were less likely to show LAER components than their NH peers (see also *Koravand et al., 2013*). Moreover, when LAER components were present, P1 and P2 were slightly later, and N2 was smaller than those of NH controls (see *Koravand et al., 2013*, for similar results regarding N2). Importantly, whereas younger (8–12 year-old) children with MMHL showed age-appropriate MMNs and LDNs to auditory oddball (large deviant) sounds, older (12–16 year-old) children with MMHL did not show MMNs, and only showed an LDN to speech. Longitudinal follow-up of a subset of children with MMHL partially replicated these

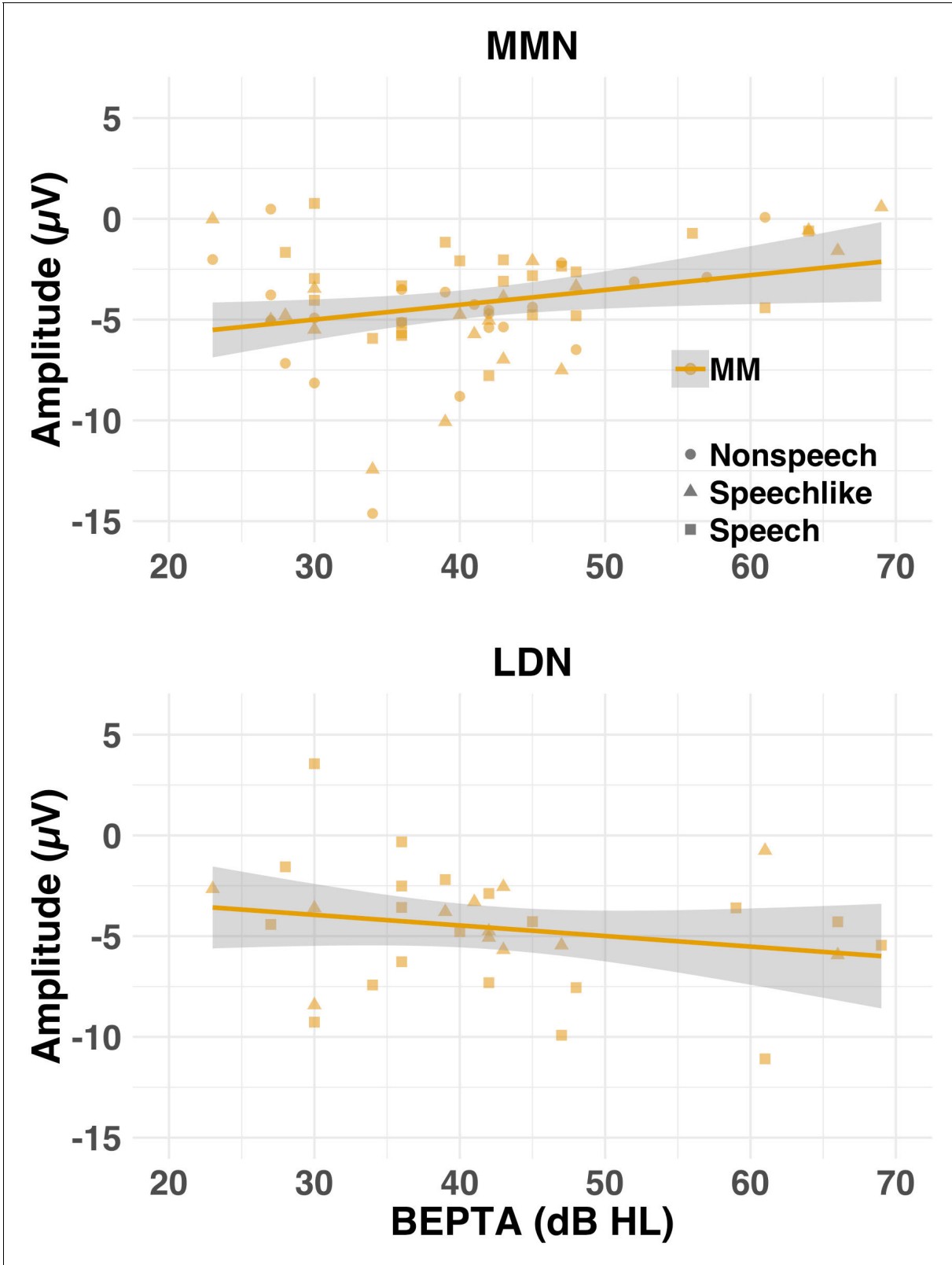

**Figure 9.** Amplitude (μV) of the MMN (top row) and LDN (bottom row) where present for the MM group as a function of severity of hearing loss (cross-sectional study). Individual (shapes) and group (lines) data are shown for each condition (nonspeech = circles; speech-like = triangles; speech = squares) and response. Shaded lines represent the 95% confidence interval. Worse BEPTA thresholds were associated with smaller MMN, but not LDN, amplitudes, where present.

*Figure 9 continued on next page*

*Figure 9 continued*

DOI: https://doi.org/10.7554/eLife.46965.014

The following figure supplement is available for figure 9:

**Figure supplement 1.** Amplitude (µV) of the MMN as a function of age of fitting with hearing aids (in months).

DOI: https://doi.org/10.7554/eLife.46965.015

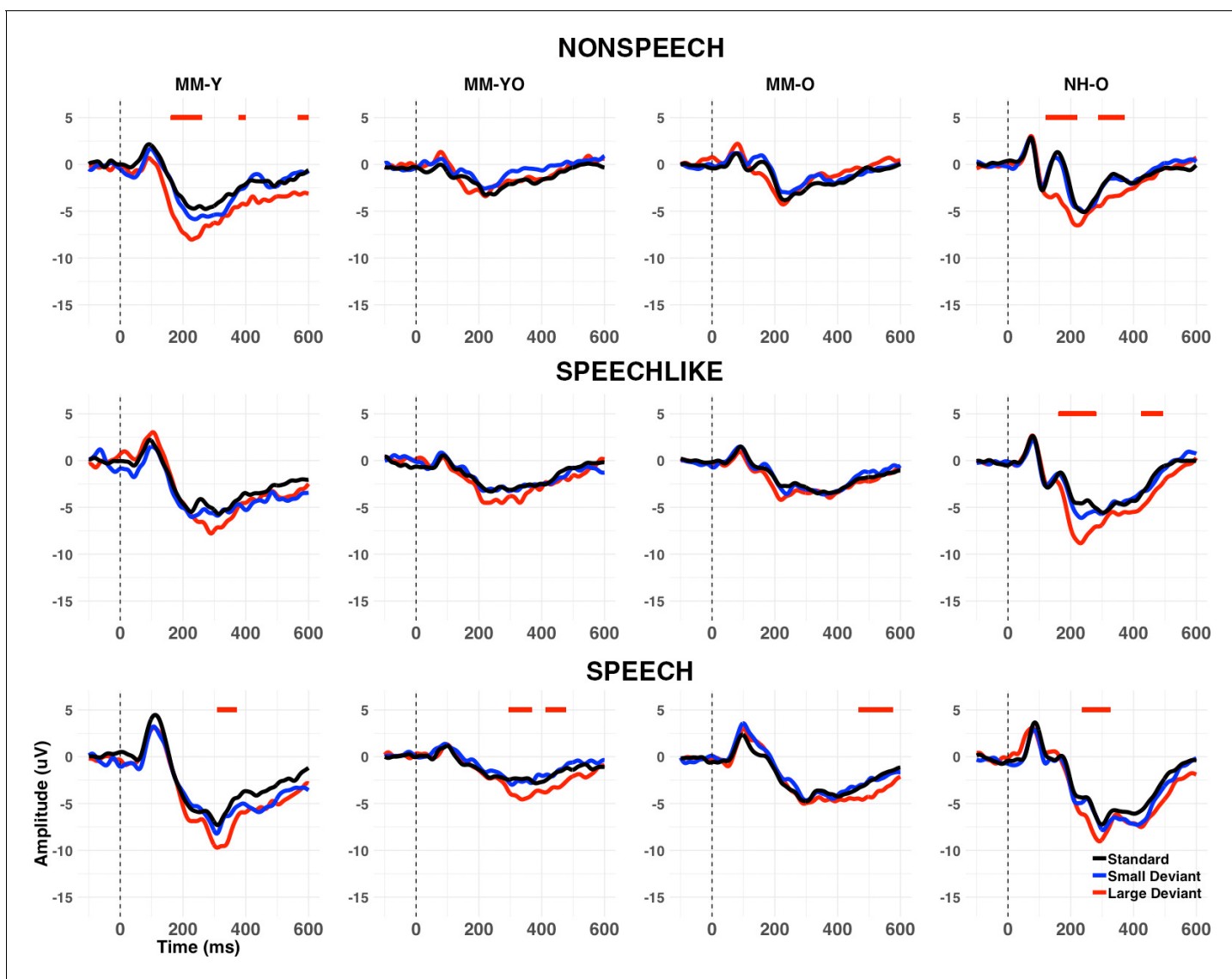

**Figure 10.** Grand average waveforms at Cz (longitudinal study). Grand averages are shown for standards (black), large deviants (red), and small deviants (blue) for each condition (nonspeech, speech-like, speech) and subgroup (MM-Y, MM-YO, MM-O, and NH-O). Note that the MM-Y plots present the grand average waveforms only for those participants from the cross-sectional study who participated in the longitudinal follow-up (n = 13); Therefore, this represents a different subset from those in *Figure 3*. Periods of the 100–400 ms (MMN) or 400–600 ms (LDN) post-stimulus epoch where a significant (p<0.01 for>20 ms consecutively) MMR was observed in the grand average are shown by horizontal lines above the waveforms, as a function of deviant type. Whereas the MM-YO subgroup obtained a significant MMN in the nonspeech and speech conditions when they were younger (MM-Y subgroup), they failed to show an MMN in the nonspeech condition six years later (MM-YO subgroup). MMN amplitude for the MM-YO subgroup was significantly smaller than that of the MM-Y subgroup across conditions, and significantly smaller than that of the NH-O subgroup (see text for details).

DOI: https://doi.org/10.7554/eLife.46965.017

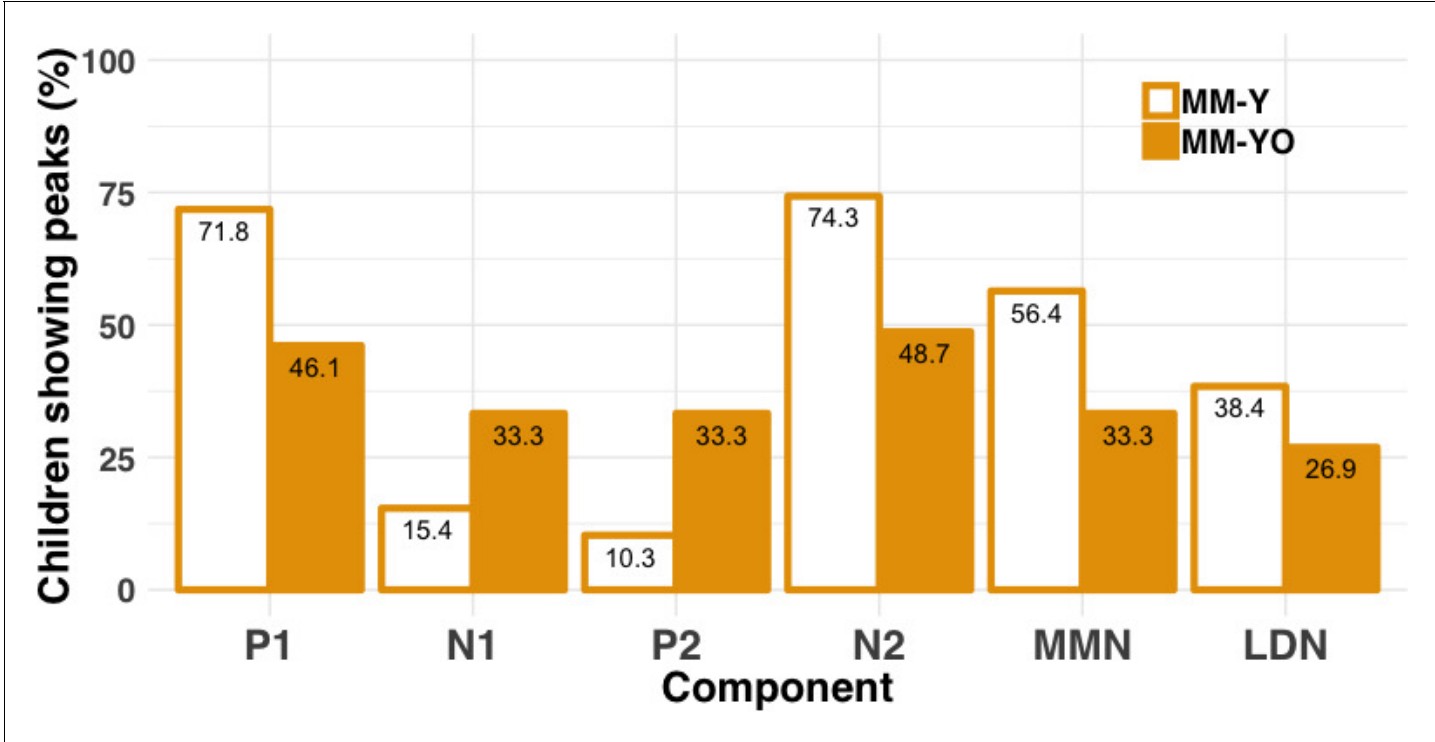

**Figure 11.** Percentages of the MM-Y and MM-YO subgroups showing present LAER components (P1, N1, P2, and N2) and MMN/LDN responses (longitudinal study). Percentages are shown for the MM-YO group at Time 1 (MM-Y) and Time 2 (MM-YO) for each component (P1, N1, P2, N2) and response (MMN and LDN). Percentages are averaged across the three conditions (nonspeech, speech-like, speech). Children in the MM-YO subgroup were less likely to show P1 or N2 components, and more likely to show P2 components, at Time 2 relative to Time 1.
DOI: https://doi.org/10.7554/eLife.46965.018

The following figure supplement is available for figure 11:

**Figure supplement 1.** Left and right ear panels: individual audiograms for each child from the MM-YO group (lines) at Time 2 (longitudinal study), with the thicker orange line indicating the group mean.
DOI: https://doi.org/10.7554/eLife.46965.019

findings; For nonspeech and speech-like stimuli, MMNs that were present at Time 1 (when children were 8–12 years old) were reduced or absent six years later (when children were 14–17 years old). However, older children with MMHL continued to show both an MMN and LDN to speech. Our results therefore demonstrate, for the first time in humans, that mild-to-moderate degradation of the auditory input during early-to-mid childhood can lead to changes in the neural processing of sounds in late childhood/adolescence.

## Four interpretations of the 'disappearing MMN'

The apparent absence of the MMN for older children with MMHL was somewhat surprising to us, hence our endeavour to replicate these findings in the form of the longitudinal follow-up. The fact that we partially did so, combined with consistent findings in the literature of reduced and delayed auditory MMNs in adults (*Oates et al., 2002*), but not in 2–12 year-old children with MMHL (*Rance et al., 2002*; *Koravand et al., 2013*; *Martinez et al., 2013*), leads us to consider four alternative explanations for our findings. First, it is possible that developmental changes to the MMN constitute neurophysiological markers for the auditory discrimination difficulties experienced by children with MMHL. The MMN is thought to reflect automatic, pre-attentive perceptual and short-term memory processes that underlie the detection of change in auditory patterns (*Näätänen, 1990*). The children with MMHL who participated in the cross-sectional study also performed more poorly than controls on a number of psychophysical auditory discrimination tasks, including those that involved stimuli almost identical to those used here (*Halliday et al., 2019*). It is therefore possible that the reduced/disappearing MMN reflected these difficulties. However, the behavioural discrimination

abilities of these children did not generally worsen with age (*Halliday et al., 2019*). Moreover, the large deviants used in this study were readily discriminable at the behavioural level by all children with MMHL except one. Finally, the perseverance of the LDN (cross-sectional study) and MMN and LDN (longitudinal study) to speech in older children with MMHL suggests that some degree of neural discrimination remained present in this group. Therefore, it seems unlikely that the absent or reduced MMN in older children with MMHL reflects the auditory discrimination difficulties of this group.

Second, it is possible that the MMN results reflected differences in the LAER components elicited by the standards. The MMN is a differential wave, and therefore its presence is dependent upon the relative morphology of the two contributing waveforms. A possibility, therefore, is that the reduction in MMN amplitude resulted from a reduction in *adaptation* in response to an ongoing stimulus (the standard), for older children with MMHL. This argument has been put forward previously, notably for N1 (see *Picton et al., 2000*; *Jääskeläinen et al., 2004*; c.f. *Garrido et al., 2009*). However, we consider it unlikely that differences in the N1 differential waveform contributed to the present results, since N1 was not present in a high proportion of children, and the time window for the MMN in this study was more consistent with the latency of the N2. Nevertheless, contrary to the adaptation hypothesis, the N2 to standards was actually *reduced* in amplitude in children with MMHL. Moreover, older children with MMHL at Time 2 were less likely to show an N2 response than their younger selves. Therefore, it seems unlikely that reduced adaptation to standards could account for the reductions in MMN amplitude seen in older children with MMHL.

## Effects of (in)audibility

A third possibility is that our results could be explained by the reduced sensation level at which children with MMHL received stimuli in the current study. LAER components and the MMN were less likely to be present in children with greater degrees of hearing loss and, where they were present, worse hearing thresholds were associated with smaller P1s and MMNs, and later N1s. Lack of audibility has been shown to result in significant changes to LAERs and MMN responses in both adults (*Oates et al., 2002*) and children (*Sharma et al., 2005*) with SNHL, as well as in NH adults (*Martin et al., 1997*; *Whiting et al., 1998*; *Martin et al., 1999*; *Billings et al., 2007*). However, in the present study, group differences in MMN amplitude were observed only for the older children with MMHL. Despite this, there were no differences in the audiometric thresholds of younger and older children with MMHL for the cross-sectional study, or between older children with MMHL and their younger selves for the longitudinal study. Unless the effects of audibility on the MMN change with age, it seems unlikely that a reduced sensation level for children with MMHL can account for our results.

## Functional changes following mild-to-moderate hearing loss

Finally, it is possible that our findings reflect developmental changes in the function and/or structure of auditory cortex that emerge during adolescence following a period of permanent, MMHL during childhood. The EEG is thought to comprise an aggregate measure of post-synaptic potentials of pyramidal cells that are oriented perpendicular to the cortical surface, forming current dipoles (*Nunez and Srinivasan, 2006*). These potentials oscillate rhythmically and synchronously within a given population of neurons in response to a stimulus, giving rise to measurable cortical ERPs. The auditory MMN has been shown to arise from the enhanced phase-synchronisation of neural oscillations in the theta (4–7 Hz) range to deviants (*Bishop and Hardiman, 2019*; *Bishop et al., 2010*), and is thought to have several generators, including bilateral temporal cortex, right inferior frontal gyrus, and bilateral frontal and centro-parietal regions (e.g. *Alho, 1995*; *Rinne et al., 2000*; *Zhang et al., 2018*). Reductions in the MMN in older children with MMHL may therefore reflect either the reduced synchronisation of a given assembly of neurons responding to the detection of a deviant stimulus or, the same degree of synchronisation but in a smaller number of coherently activated neurons. However, these changes appear to relate to deviant stimuli only, and emerge with age; Normal cortical phase-synchrony to speech standards (/ba/) has been reported for young (Median age ~2.7 years) children with MMHL (*Nash-Kille and Sharma, 2014*). The fact that reductions in the MMN are seen only for older children with MMHL likely reflects the prolonged maturational time-course of this response, which has been shown to extend into adolescence (*Bishop et al., 2011*).

This explanation gains support from recent studies in gerbils that have shown that mild or moderate levels of hearing loss during early life can induce lasting changes to auditory cortical development (see *Sanes, 2016* for a review). *Takesian et al. (2012)* found that early-induced bilateral mild-to-moderate conductive hearing loss led to a reduction in inhibitory synaptic strength in thalamo-cortical brain regions that persisted into adulthood, along with delays in the maturation of synaptic decay time. *Mowery et al. (2015)* showed that even brief periods of mild, bilateral conductive hearing loss led to lasting changes in the membrane, firing properties, and inhibitory synaptic currents of auditory cortical pyramidal neurons. However, these changes only persisted if the auditory deprivation occurred during a discrete critical period (*Mowery et al., 2015*). More recently, *Mowery et al. (2017)* reported that transient hearing loss caused an immediate imbalance of excitatory and inhibitory gain and reduced firing rate in auditory cortical neurons. Although these changes were reversed by the restoration of hearing in early life, adult gerbils continued to show such changes to neurons within the dorsal striatum, even after hearing was restored (*Mowery et al., 2017*).

With regards to the present study, it is not clear whether the changes observed were attributable to the fact that the children with MMHL received intervention that was late (i.e., after the critical period), inconsistent, or inadequate. In humans with MMHL, a period of auditory deprivation is likely, especially for those with mild SNHL, who are currently not routinely detected by newborn hearing screening programmes (*Bamford et al., 2007*; *Carew et al., 2018*). Indeed, for the present study, age of detection ranged from 2 months to 14 years (M ~ 4.5 years), and age of hearing aid fitting from 3 months to 15 years (M ~ 5.4 years). Nevertheless, age of hearing aid fitting did not correlate with MMN amplitude in children with MMHL after controlling for severity of hearing loss (see *Figure 9—figure supplement 1*). It is also possible that children with MMHL did not use their hearing aids consistently, leading to suboptimal outcomes (*Scherer, 1996*; *Walker et al., 2013*). Although we did not measure daily hearing aid use in the current study, we observed that for the cross-sectional study, three of the older children with MMHL had not been prescribed with hearing aids, and two were refusing to wear their hearing aids (see also *Scherer, 1996*). All of the children in the longitudinal study had been prescribed with hearing aids; However, we do not know how consistently they used them. Finally, even when children with MMHL do receive early and consistent remediation, hearing aids are unable to redress many of the perceptual consequences of SNHL, such as the broadening of auditory filters and changes in sensitivity to temporal fine structure and envelope cues (see *Halliday et al., 2019*). This differs from the transient conductive hearing loss induced in animal studies, where hearing sensation is restored following earplug removal (*Takesian et al., 2012*; *Mowery et al., 2015*; *Mowery et al., 2017*). Further research is needed to determine whether children with MMHL would benefit from earlier detection and intervention (in the form of hearing aids c.f. *Carew et al., 2018*), more consistent use of hearing aids, and/or improvements in signal quality, via enhancements to hearing aid processors.

## Speech as 'special'?

A final finding of our study was that of a preserved LDN for the cross-sectional study, and both MMN and LDN for the longitudinal study, to speech stimuli for older children with MMHL. It is possible that these findings were due to differences in the spectral composition between the different stimuli (see *Figure 2*). For the nonspeech condition, discrimination between the standard and the large deviant depended on information contained within a very narrow region (of about 200 Hz) around 1 kHz. For the speech-like condition the relevant region was wider (between around 1–2 kHz). For the speech condition differences between the /ba/ standard and /da/ large deviant occurred across 0.5–4 kHz, although discrimination was likely to depend upon differences in the properties of the initial burst, along with the subsequent formant transitions. Better performance in the speech condition may therefore be explained in terms of the broader frequency range over which changes could be detected. An alternative possibility is that these findings reflect an increasing specialisation of auditory cortex to speech with age for children with MMHL. Consistent with this, the MM group from the cross-sectional study showed improvements in their behavioural speech discrimination thresholds with age (*Halliday et al., 2019*). Finally, a third possibility is that these findings were also a consequence of decreases in neural oscillatory synchronisation with age for children with MMHL. Contrary to the MMN, the LDN has been shown to reflect increased *desynchronization* of neural oscillations to deviants across delta, theta and alpha bands (*Bishop et al., 2010*). The LDN is more likely to be present for speech than nonspeech stimuli (*Korpilahti et al., 2001*), and is more

prominent for children than adults (*Kraus et al., 1993*; *Cheour et al., 2001*). It has been argued that increases in cortical desynchronization reflect either a decrease in neural synchrony amongst underlying neurons (*Pfurtscheller and Lopes da Silva, 1999*), or the same synchrony but over a more focal region (*Bishop et al., 2010*). Reductions in neural inhibition have been shown to lead to increases in the variability of timings of neural oscillations, and a consequent loss of neural synchrony (*Isaacson and Scanziani, 2011*). If the neurons of children with MMHL show reduced inhibition (as do those of gerbils with conductive hearing loss; *Takesian et al., 2012*; *Mowery et al., 2015*; *Mowery et al., 2017*), this would be expected to lead to reductions in neural synchronisation (i.e., the MMN) following exposure to deviant stimuli. However, neural desynchronization (i.e., the LDN) may be expected to remain intact. Further research is needed to examine changes to the neural oscillations of children with MMHL in mid-to-late childhood.

## Conclusion

In conclusion, our results provide evidence for functional changes in the neural processing of auditory stimuli during adolescence following early, childhood mild-to-moderate SNHL. These changes appear to manifest as a reduction in the MMN, a neural signature for change detection in auditory signals. It is possible that these changes may impede the extraction of regularities in the speech signal that are important for language-learning, perhaps accounting for the higher-than-expected presence of language difficulties in children with MMHL (*Halliday et al., 2017a*). Earlier detection and treatment of MMHL may go some way towards mitigating the effects of MMHL on the developing auditory system in children.

# Materials and methods

## Cross-sectional study (Time 1)

### Participants

Participants were recruited as part of a larger study, which entailed psychophysical and electrophysiological measures of auditory processing, as well as psychometric assessments of language and cognitive functioning (see *Halliday et al., 2017a*; *Halliday et al., 2017b*). The study was conducted with the verbal assent of the participants and the written informed consent of their parents/guardians, and was approved by the UCL Research Ethics Committee. Unidentifiable data is available on GitHub (*Calcus, 2019*; copy archived at https://github.com/elifesciences-publications/MMHL).

Children with MMHL were approached via Hearing Services in Local Educational Authorities across Greater London and the South East of England. Inclusion criteria were: (a) a diagnosis of MMHL, defined as a BEPTA threshold of 21–40 dB HL (mild) or 41–70 dB HL (moderate) across octave frequencies 0.25–4 kHz (*British Society of Audiology, 2011*), (b) being 8–16 years old at the time of testing, (c) a monolingual English-speaking background, and (d) communicating solely via the oral/aural modality (i.e. non-signers). Exclusion criteria were (a) any known medical, neurological or psychological conditions other than SNHL, and (b) SNHL that could be attributed to a syndrome, neurological impairment (including Auditory Neuropathy Spectrum Disorder; ANSD), or a known post-natal event (e.g. measles). Children who met the inclusion criteria and did not meet the exclusion criteria (n = 57) were then invited into UCL to attend a screening session. During this session, air-conduction BEPTA thresholds across octave frequencies from 0.25 to 8 kHz were verified using an Interacoustics AC33 audiometer according to recommended procedures (*British Society of Audiology, 2011*). Information regarding children's medical and audiological histories was collected via an in-house parent questionnaire. In addition, nonverbal IQ was assessed using the Block Design subtest of the Wechsler Abbreviated Scale of Intelligence (*Wechsler, 1999*). Children who failed to obtain an air-conduction BEPTA threshold equivalent to a MMHL, or a nonverbal IQ T-score of at least 40 (i.e. equivalent to an IQ standard score of 85), were excluded (n = 5). Moreover, those who failed to participate in all conditions of the study (n = 6) were also excluded. This left a final sample size of 46 children (19 mild, 27 moderate) for the MM group. A total of 43 had been prescribed with bilateral hearing aids, which were worn only during the psychometric assessments and some of the psychophysical assessments (see *Halliday et al., 2019*), although two participants were refusing to use their aids. Age of confirmation of SNHL ranged from 2 months to 14 years (Median (Mdn) = 57

months, M = 54 months, SD = 36), and age of hearing aid fitting from 3 months to 15.6 years (Mdn = 64.5 months, M = 63.1 months, SD = 39.6). Audiograms are displayed in *Figure 1*.

Children in the MM group were individually matched in age (±6 months) to NH children from schools in the same geographical location (NH group; n = 44). In two instances, children from the MM group were matched to a single child from the NH group. No child in the NH group had a known history of hearing loss (including otitis media with effusion), educational difficulties, or speech and language problems (based on parent/guardian report). All achieved mean pure-tone-average (PTA) air-conduction thresholds of ≤20 dB HL, across octave frequencies 0.25–4 kHz in both ears, and obtained thresholds ≤ 25 dB HL at all octave frequencies from 0.25 to 8 kHz (see *Figure 1*). All achieved a nonverbal IQ T-score (Block Design) of ≥40 (*Wechsler, 1999*).

In order to examine the developmental effects of MMHL on the LAER, MMN and LDN, both groups were further divided into two age bands based on a median split for the MM group (12.0 years), and roughly corresponding to known changes in the morphology of the LAER in children of this age range (*Bishop et al., 2007*). *Younger* (Y) children were aged 8–11 years, and *Older* (O) children were aged 12–16 years (see *Table 1*). Participant characteristics of the two groups (MM vs NH) as a function of age band (Y vs O) are presented in *Table 1*. A series of univariate analyses of variance (ANOVA) was conducted with group and age band as the two between-subjects variables and the group ×age band interaction included in the initial models, the latter being subsequently removed in instances where it was non-significant (this was true in all instances). Significance values were adjusted for multiple comparisons (α = 0.016). As expected, the MM group did not differ from the NH group in age [$F(1, 85)$=3.02, p=0.086, $\eta^2$ = 0.03] but had poorer BEPTA thresholds [$F(1, 85)$ =341.35, p<0.001, $\eta^2$ = 0.80]. However, the age at which children's mothers' left full-time education (*maternal education*; a measure of socio-economic status) just missed significance [$F(1, 81)$=3.45, p=0.066, $\eta^2$ = 0.04]. Unexpectedly, the MM group also had marginally lower nonverbal IQ relative to NH controls [$F(1, 85)$=9.68, p=0.003, $\eta^2$ = 0.10]. However, this was driven by the higher-than-expected nonverbal IQ of the NH group; The nonverbal IQ scores of the MM group were all within the normal range, and the mean T-score for this group (M = 55.6, SD = 8.7) was higher than the normative mean (M = 50; SD = 10; see *Table 1*). Across groups, children in the Y and O age bands did not differ in maternal education [$F(1, 81)$=0.21, p=0.870, $\eta^2$ = 0.00], nonverbal IQ [$F(1, 85)$=2.41, p=0.124, $\eta^2$ = 0.03], or BEPTA thresholds [$F(1, 85)$=0.31, p=0.579, $\eta^2$ <0.01].

Because we found differences in the neural responses of our MM-Y and MM-O subgroups, we also examined whether these subgroups differed in their audiological, cognitive, or demographic profiles. Children in the MM-Y and MM-O subgroups did not differ from each other in their BEPTA thresholds, [$F(1, 44)$=0.18, p=0.672, $\eta^2$ = 0.00], worse-ear pure-tone-average (WEPTA) thresholds [$F(1, 44)$=1.27, p=0.264, $\eta^2$ = 0.04], or mean (both ears) PTA (MePTA) audiometric thresholds [$F(1, 44)$ =0.49, p=0.486, $\eta^2$ = 0.01]. Likewise, they did not differ in their age of detection of SNHL [$t(43)$ =−0.85, p=0.400, $d$ = 0.25] or age at which hearing aids were first prescribed [$t(12.6)$=0.56, p=0.582, $d$ = 0.25]. However, a larger proportion of older than younger children did not have (n = 3 for MM-O subgroup) or did not wear (n = 2 for MM-O subgroup) hearing aids [Fisher's Exact test, two-sided p=0.049]. Finally, children in the MM-Y subgroup had marginally but not significantly higher nonverbal IQ than those in the MM-O subgroup [$F(1, 40)$=3.90, p=0.055, $\eta^2$ = 0.09], but maternal education levels did not differ between groups [$F(1, 40)$=0.00, p>0.10, $\eta^2$ = 0.00].

## Procedure

A passive, two-deviant oddball paradigm (standard probability: 70%; small and large deviant probability: 15% each) was used. There were three conditions (nonspeech, speech-like, and speech), each divided into two sessions lasting approximately 5 min with a short break in between, leading to a total recording time of around 30 min (without breaks). Each of the conditions consisted of 660 stimuli presented with a stimulus-onset-asynchrony of 1 s. The order of conditions was counter-balanced across groups and age bands. During the recording, participants were comfortably seated in an electrically shielded, sound-attenuated booth. Stimuli were presented using Presentation software (Neurobehavioural systems, version 14.8.1) at a constant intensity of 70 dB SPL via insert earphones (Etymotic ER-2). Stimuli were presented to all children at the same level, and without any of the MM group using hearing aids. Participants watched a silent DVD during the recording.

## Stimuli

Stimuli were taken from three continua, constructed for the psychophysical tests that were undertaken as part of the wider study (see *Halliday et al., 2017a*; *Halliday et al., 2019*). Stimuli are available on GitHub (*Calcus, 2019*; copy archived at https://github.com/elifesciences-publications/MMHL). Stimuli were (a) sinusoidally frequency-modulated pure tones (nonspeech), (b) complex periodic tones with a vowel-like formant structure whose second formant was sinusoidally frequency-modulated (speech-like), and (c) consonant-vowel (CV) speech syllables (speech; see *Figure 2*). All stimuli were 175 ms in duration, and both nonspeech and speech-like stimuli had a 15 ms linear on-off ramp. For all conditions, the difference between both the large deviant and the standard and the small deviant and the standard was clearly detectable during pilot testing (NH child listeners, n = 4). In addition, the difference between the large deviant and the standard was detectable during psychophysical testing by all but one child with MMHL in the cross-sectional study (the one with the highest BEPTA threshold; see *Halliday et al., 2019*).

For the nonspeech condition, one end of the continuum (the standard) was a 1-kHz pure tone, and the other was a sinusoid frequency-modulated at a rate of 40 Hz and a modulation depth of 10% (the large deviant; corresponding to a modulation index of 1). A continuum of 98 frequency-modulated stimuli was created between these two endpoints, all at a rate of 40 Hz, and ranging from a modulation index of 0.02 in 0.01 steps. The small deviant was chosen from these 98 stimuli, and had with a modulation depth of 1% (corresponding to a modulation index of 0.1). The initial phases of both the standard (non-modulated) and deviant (modulated) stimuli were 0°.

For the speech-like condition, stimuli were complex harmonic sounds, containing 50 equal-amplitude harmonics passed through three simple resonators. These resonators all had a bandwidth of 100 Hz, and centre frequencies of 500, 1500 and 2500 Hz, leading to an overall spectrum with three formants, characteristic of a neutral vowel (/ə/). The second formant (F2) was unmodulated for the standard. For the deviant stimuli, a continuum of 100 stimuli was created, with F2 modulated at a rate of 5.714 Hz, and the frequency deviation ranging from ±1 Hz to - ±200 Hz (the large deviant), spaced logarithmically. The small deviant had a frequency deviation of 20 Hz.

For the speech condition, the standard was a digitised /ba/ syllable and the large deviant was a digitised /da/ syllable. Both were spoken by a female speaker and identical to those used in *Bishop et al. (2010)*. Consonant burst-time differences for these stimuli were minimised, with the intonation contours being equated using Praat (*Boersma and Weering, 2005*) and the final stimuli being RMS equalised with Gold-Wave (*Craig, 2008*). Thus, consonant change detection was primarily based on formant transitions into the vowel. A continuum of 98 stimuli was created in between the two endpoints, using the morphing capabilities of the programme STRAIGHT (*Kawahara et al., 1999*). This programme uses pitch-adaptive spectral analysis including fundamental frequency extraction, combined with a surface reconstruction method to generate a smooth trajectory of high-quality stimuli. The small deviant was stimulus # 50 on the continuum (i.e. in the middle) and was perceived by the authors as sounding /ga/-like.

## Data recording and analysis

EEG activity during audio presentation was recorded using a NuAmps 40-channel monopolar digital amplifier and NeuroScan 4.4 Acquire software system (Neuromedical Supplies) at a sampling rate of 500 Hz. The EEG was recorded from 28 scalp electrodes, positioned in the standard 10/20 configuration. Additional electrodes were placed on the mastoids, and above and below the left eye (Vertical EOG) and on the outer canthi of both eyes (Horizontal EOG) to monitor eye movements. Skin-to-electrode impedances were below 5 kΩ at the start of the recording and monitored to ensure they remained below 5 kΩ. The EEG signal was amplified with a gain of 20000 and band-pass filtered at 0.1–70 Hz.

EEG data were analysed offline using the EEGLAB toolbox (*Delorme and Makeig, 2004*) and custom Matlab software. The EEG was filtered using an FIR band-pass filter at 0.5–35 Hz, and re-referenced to the averaged mastoids. Data were epoched relative to stimulus onset using a −200 to 800 ms window and baseline corrected from −200 to 0 ms. Epochs containing artefacts were rejected (mean number rejected epochs across groups, age bands and conditions: 8.9 standards, 3.5 small deviants, 3.6 large deviants) from the final averages on the basis of a ± 100 μV criterion for all

channels. Individual epochs were averaged separately for each of the three conditions, and for standards and deviants. Standards directly following a deviant were excluded from the average.

## Identification of LAER components

For each participant, the LAER components P1, N1, P2, and N2 at Cz that were evoked by the standards for each condition were identified by two independent judges (including author OT). A third judge (author AC) made the final decision where there was disagreement between judges. All judges were experienced with clinical and research developmental EEG data. Cz was selected because it was where responses were largest overall, as is typical for this sort of data (*Ponton et al., 2000a*). Anonymized individual waveforms for each condition were presented to the judges, together with the grand-average NH waveform observed for the same condition for both the Y and O subgroups as a comparison point. Judges were blind to the participant group, age band, and stimulus condition for each waveform. Judges were required to indicate the presence/absence of each peak and, if present, their latency. For a response to be considered present, at least two out of the three judges had to agree on the peak latency of each component in the individual waveform. Consistency across the three judges reached 82% in the nonspeech condition, 74% in the speech-like condition, and 81% in the speech condition (consistency in line with those observed by *Oates and Stapells, 1997*).

## Identification of the MMN and LDN

Voltage maps confirmed that the MMN was strongest in the fronto-central region (at around 200 ms post-stimulus onset, see *Figure 3*), as is typical for this response (*Näätänen et al., 2007*). Statistical analyses for the MMN and LDN were therefore conducted on data recorded at electrode Fz. To calculate the MMN/LDN, the responses evoked by standards were subtracted from responses evoked by deviants for each participant for each condition. Differential waves were obtained separately for the small and large deviants.

The presence of a significant MMN or LDN was assessed in two ways. First, for each group and age band, point-to-point comparison of the differential wave amplitudes was performed in order to determine the latency period over which the waveforms were significantly different from zero, if any. One-sided $t$-tests were computed within the 100–400 ms post-stimulus-onset time window - identified as the region most likely to contain the MMN by visual inspection of the individual waveforms - with a sampling rate of 500 Hz. The 400–600 ms time window was selected for the LDN. Because adjacent points in the waveform are highly correlated potentially leading to spurious significant values in short intervals, an MMN/LDN was considered present when p<0.01 (one-tailed) for >20 ms at adjacent time-points (*Kraus et al., 1993*; *McGee et al., 1997*). Next, this process was repeated at the individual level using the grand average waveforms obtained for each group, age band, and condition, with an $\alpha$ level <0.05 (one-tailed) for >20 ms at adjacent time-points.

Amplitude of the MMN/LDN was calculated from the differential waveforms for each participant, condition, and deviant type. Amplitude of the MMN was defined as the mean amplitude (in µV) in a 50-ms window centred around the NH-group grand-average peak latency for the appropriate age band, condition, and deviant type (see *Appendix 1—table 9*). Unlike the MMN, there was no clear overlap in the periods of significant LDN between groups of corresponding age bands (see *Appendix 1—table 2*). Therefore, amplitude of the LDN was defined as the mean amplitude (in µV) in a 50-ms window centred around each group's own grand-average peak latency in the given age band, condition and deviant type.

## Statistical analyses

A series of mixed-effects regression models was applied using the lme4 package of R (*Bates et al., 2014*). Two sorts of analyses were conducted. Logistic mixed-effects regressions were conducted to determine whether group (NH vs. MM), age band (Y vs. O), condition (nonspeech, speech-like, and speech), component (P1, N1, P2, N2) or response (MMN or LDN) predicted the presence of a component/response. Linear mixed-effects regressions were conducted to determine whether those same independent variables predicted the amplitude (P1-N2, MMN, and LDN) or latency (P1-N2 only) of these components/responses. Starting from saturated models with listener as random intercepts, group, age band, condition, and component as main effects, and all two-, three- and four-way

interactions included (which were treated as fixed-effect terms), predictors were removed iteratively if their removal did not significantly worsen the fit of each model. The most parsimonious models were selected using the Akaike Information Criterion (AIC). Final models included only significant main effects and/or interactions, and Pseudo $R^2$ (Nagelkerke) was used as an index of goodness-of-fit. Post-hoc analyses were $t$-tests, and results were reported after correction for multiple comparisons whenever necessary.

## Longitudinal study (Time 2)

### Participants

Six years later (Time 2), we attempted to re-contact all of the 23 children who initially constituted the MM-Y group from the cross-sectional study (at Time 1) and who had given their permission to be re-contacted at a future point in time. Fourteen of these children were successfully re-contacted, and all were re-tested 6 years after their initial participation (Time 2). These children therefore constituted a relatively random sample from the original MM-Y subgroup of the cross-sectional study. Results from one child could not be included as their hearing had significantly deteriorated since Time 1. The remaining children (n = 13) comprised the MM-YO group for the longitudinal study.

Results from the MM-YO group were compared to those of three subgroups taken from the cross-sectional study: the MM-Y subgroup (i.e. themselves, 6 years previously), the MM-O subgroup (i.e. a different group of children with MMHL of a similar age), and the NH-O group (i.e. NH controls of a similar age). Note that the MM-Y group at Time 2 was created by selecting the subset of 13 children from Time 1 who also took part at Time 2. The complete sample from the original MM-O (n = 23) and NH-O (n = 18) subgroups were included as comparisons to maximise power.

Participant characteristics of the four subgroups (MM-Y, MM-YO, NH-O, MM-O) are presented in *Table 2*. Univariate ANOVA showed a main effect of subgroup on age [$F$(2, 85)=7.93, p<0.001, $\eta^2$ = .23]; Owing to the time lag, the MM-YO subgroup was slightly older than both the MM-O and NH-O subgroups (on average by 1.4 years) [$t$(34)=3.92, p<0.001, $d$ = 1.31, CI (.78, 2.45), and $t$(29) = 2.65, p=0.013, $d$ = 0.92, CI (0.29, 2.22), respectively]. Age was therefore used as a covariate in the analyses for the longitudinal study. The MM-YO subgroup did not differ from the MM-O or NH-O subgroups in nonverbal IQ [$t$(19.4)=−2.05, p=0.054, $d$ = −0.77, and $t$(17.6) = −0.44, p=0.664, $d$ = −0.17, respectively], or maternal education [$t$(24.8)=0.18, p=0.855, $d$ = 0.06, and $t$(27.9) = −1.21, p=0.233, $d$ = −0.43, respectively]. The MM-YO subgroup also did not differ in their BEPTA, WEPTA, or MePTA audiometric thresholds relative to the MM-Y subgroup at Time 1 [$F$(3, 22)=0.675, p=0.576], but did differ from the MM-O subgroup [$F$(3, 32)=3.22, p=0.035] (MANOVAs, see *Figure 11—figure supplement 1* and *Table 2*). This was due to the MM-YO subgroup having marginally but non-significantly better BEPTA thresholds than the MM-O subgroup from Time 1 [$t$(32.7)=1.83, p=0.076, $d$ = 0.572]. The MM-YO subgroup did not differ from the MM-O subgroup in age of detection [$F$(1, 19)=0.01, p=0.975] age of hearing aid fitting [$F$(1, 19)=0.09, p=0.756], or in the proportion of children wearing hearing aids [Fisher's Exact test, two-sided p=0.136].

### Data recording and analysis

The experimental setup for stimulus presentation and EEG recording was identical to that used at Time 1, with two main differences. First, as the system was older, technical difficulties led to us to discard EEG activity recorded on the left mastoids. Therefore, data from the MM-YO subgroup was only re-referenced to the right mastoids. Second, some scalp channels appeared consistently faulty, hence increasing the noise floor at Time 2. Because the MMN typically displayed a fronto-central distribution for the cross-sectional study, we thus limited the artefact rejection to a threshold of ±100 μV criterion in either Cz or Fz. Data from the three other subgroups (MM-Y, MM-O, NH-O) were thus reprocessed using the same re-referencing and artefact rejection criterion as those applied to the MM-YO subgroup.

The numbers of rejected standard trials in each condition were compared between the MM-Y and MM-YO subgroups (i.e. within-subject) as an index of noise floor in the two studies. There was no significant difference in noise floor between studies in any of the three conditions [nonspeech: $\chi^2$(12)=13.3, p=0.420; speech-like: $\chi^2$(12)=7.73, p=0.800; speech: $\chi^2$(12)=20.3, p=0.120].

## Acknowledgements

The authors would like to thank all the children who participated in this study and their parents. We gratefully acknowledge funding from the Economic and Social Research Council (RES-061-25-0440) and the People Programme (Marie Curie Actions) of the European Union's Seventh Framework Programme FP7/2007–2013/under REA grant agreement n° FP7-607139 (iCARE). Thanks go to Steve Nevard and Andrew Clark for their technical help with this project, and to Páraic Scanlon and Xiao-wen Wang for their help in recruiting and testing the children.

## Additional information

### Funding

| Funder | Grant reference number | Author |
|---|---|---|
| H2020 Marie Skłodowska-Curie Actions | FP7-607139 | Lorna F Halliday |
| ESRC | RES-061-25-0440 | Lorna F Halliday |

The funders had no role in study design, data collection and interpretation, or the decision to submit the work for publication.

### Author contributions

Axelle Calcus, Data curation, Formal analysis, Visualization, Writing—original draft, Writing—review and editing; Outi Tuomainen, Data curation, Supervision, Investigation, Writing—review and editing; Ana Campos, Investigation, Writing—review and editing; Stuart Rosen, Software, Formal analysis, Methodology, Writing—review and editing; Lorna F Halliday, Conceptualization, Resources, Data curation, Software, Formal analysis, Supervision, Funding acquisition, Validation, Investigation, Methodology, Writing—original draft, Project administration, Writing—review and editing

### Author ORCIDs

Axelle Calcus https://orcid.org/0000-0002-1240-1122
Lorna F Halliday https://orcid.org/0000-0003-1883-7741

### Ethics

Human subjects: Informed consent, and consent to publish was obtained from parents/guardians of the children included in this study. Ethical approval for this study was provided by the UCL Research Ethics Committee (Project ID number: 2109/004).

### Decision letter and Author response

Decision letter https://doi.org/10.7554/eLife.46965.032
Author response https://doi.org/10.7554/eLife.46965.033

## Additional files

### Data availability

Unidentifiable data, stimuli, and statistical analyses scripts are available on https://github.com/acal-cus/MMHL (copy archived at https://github.com/elifesciences-publications/MMHL).

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

# Appendix 1

DOI: https://doi.org/10.7554/eLife.46965.020

## Cross-sectional study (Time 1)

### Modelling Results

#### Late Auditory Evoked Responses

A series of mixed-effects logistic regression models was applied using the lme4 package of R (**Bates et al., 2014**) to determine whether Group, Age band, Condition, or ERP Component predicted the presence/absence of a peak (see *Appendix 1—table 1*). Starting from a saturated model with Listener as a random intercept, Group, Age band, Condition, and Component as main effects, and all two-, three- and four-way interactions included (which were treated as fixed-effect terms), predictors were removed iteratively if their removal did not significantly worsen the fit of the model. The best fitting model [$\chi^2(8)=313.29$, p<0.001, *AIC* = 1181.0, $R^2_c=0.335$] included the main effects of Group, $\chi^2(1)=96.6$, p<0.001, $R^2_m$ = .125, Age band, $\chi^2(1)=2.60$, p=0.106, $R^2_m$ = .014, and Component, $\chi^2(3)=225.8$, p<0.001, $R^2_m$ = .250, as well as an Age band ×Component interaction, $\chi^2(3)=9.58$, p=0.022, $R^2_m$ = .011, although the main effect of Age band was not significant. Children from the MM group were less likely to have present components than children from the NH group [odds ratio (OR): 0.24, p<1e$^{-15}$]. The main effects of Age band and Component have to be considered in light of their interaction. Post-hoc logistic regressions conducted separately on each component revealed that for P1, N1 and N2, there was no significant effect of Age band on presence/absence, OR: 0.73, p=0.282, OR: 1.38, p=0.202, and OR: 0.66, p=0.174, respectively. However, P2, Age band was a significant predictor of component presence, OR: 1.71, p=0.038. Regardless of Group, P2 were more likely to occur in the O than the Y Age bands.

**Appendix 1—table 1.** Results of the logistic regression analyses for presence of LAER and MMRs components (cross-sectional study).

| Component | Effects | $\chi^2$ | Df | p | $R^2_m$ |
|---|---|---|---|---|---|
| LAERs | Age band | 2.60 | 1 | .106 | .014 |
| | Group | 96.6 | 1 | < .001 | .125 |
| | Component | 225.8 | 3 | < .001 | .250 |
| | Age band × Component | 9.58 | 3 | .022 | .011 |
| MMN | Group | 7.39 | 1 | .007 | .033 |
| LDN | Group | 1.33 | 1 | .247 | .007 |

*Note.* The best fitting models for LAERs and MMRs presence were all an acceptable fit (LAERs [AIC = 1181, $R^2_c=0.335$]; MMN [AIC = 369.4, $R^2_c=0.033$]; LDN [AIC = 243.2, $R^2_c=0.009$]). Effects that were significant are shown in boldface.

DOI: https://doi.org/10.7554/eLife.46965.021

**Appendix 1—table 2.** Periods (>20 ms) of the 100–600 ms post-stimulus epoch (in ms) that contained a significant MMN (100–400 ms) or LDN (400–600 ms) for each Condition (nonspeech, speech-like, speech), Deviant type (small and large), Group (NH and MM), and Age band (Y and O) (cross-sectional study)

| Condition | Deviant | Younger (Y) | | Older (O) | |
| | | NH-Y (n = 26) | MM-Y (n = 23) | NH-O (n = 18) | MM-O (n = 23) |
|---|---|---|---|---|---|
| Nonspeech | Small | - | 228–262 | - | - |
| | Large | 126–230 | 154–258 | 120–210 322–348 | - |
| Speech-like | Small | 252–278 | - | - | - |
| | Large | 212–302 400–442 | 242–370 564–592 | 172–284 408–470 | - |
| Speech | Small | - | - | - | - |
| | Large | 226–344 404–546 | 204–242 272–548 | 230–298 428–462 490–522 | 500–580 |

*Note.* Presence/absence of the MMR was determined through point-to-point comparison of the differential wave amplitudes to calculate the latency period over which the waveforms were significantly different from zero. Unilateral t-tests were computed within the 100–400 and 400–600 ms post-stimulus-onset time windows with a sampling rate of 500 Hz. An MMN or LDN was considered present when p<0.01 for>20 ms at adjacent time-points (see text).

DOI: https://doi.org/10.7554/eLife.46965.022

**Appendix 1—table 3.** Results of the linear mixed-effect analyses for the amplitude of MMN and LDN (cross-sectional study).

| Component | Effects | $\chi^2$ | *Df* | *p* | $R^2_m$ |
|---|---|---|---|---|---|
| MMN | Group | 14.48 | 1 | < .001 | .080 |
| | Age band | 5.31 | 1 | .021 | .036 |
| | Condition | 7.73 | 2 | .021 | .033 |
| | Age band × Group | 3.29 | 1 | .069 | .013 |
| LDN | Group | 1.95 | 1 | .162 | .010 |

*Note.* The best fitting models for MMN and LDN amplitude were both an acceptable fit (MMN [AIC = 1418.1, $R^2_c$ 0.27]; LDN [AIC = 933.4, $R^2_c$ 0.010]).

DOI: https://doi.org/10.7554/eLife.46965.023

A series of linear mixed-effects models was conducted (lme4 package of R; **Bates et al., 2014**) to determine whether Group, Age band (Y vs. O), or Condition predicted the latencies and/or amplitudes of the P1, N1, P2, and N2 components, where present (see **Appendix 1—table 4**). The same backward fitting strategy was used as for the logistic regression. Conditional R squared ($R^2_c$) from the MuMIn package of R is reported as an index of goodness-of-fit (**Nakagawa and Schielzeth, 2013**), as it includes variance explained by both fixed and random factors. Additionally, effect sizes for individual predictors were computed as the difference in marginal R squared ($R^2_m$) for a model with and a model without that particular predictor. Separate models were fitted for latency and amplitude and for each component. Note that these analyses represent the 'best case scenario' for children with MMHL, in that only those children who showed these components were included in the analyses.

**Appendix 1—table 4.** Results of the linear mixed-effect analyses for the amplitude and latency of each LAER component (P1, N1, P2, N2), where present (cross-sectional study).

| Component | Effects | Amplitude $\chi^2$ | Df | p | $R^2_m$ | Latency Effects | $\chi^2$ | Df | p | $R^2_m$ |
|---|---|---|---|---|---|---|---|---|---|---|
| P1 | **Condition** | 22.78 | 2 | <0.001 | .058 | **Group** | 15.29 | 1 | < .001 | .055 |
|  |  |  |  |  |  | **Condition** | 6.32 | 2 | .012 | .037 |
| N1 | Age band | 0.97 | 1 | .324 | .011 | **Age band** | 7.87 | 1 | .005 | .057 |
|  | **Condition** | 6.67 | 2 | .035 | .048 | **Condition** | 19.88 | 2 | < .001 | .157 |
|  | **Age band × Condition** | 9.49 | 2 | .008 | .063 |  |  |  |  |  |
| P2 | **Age band** | 7.07 | 1 | .008 | .103 | Group | 3.80 | 1 | .051 | .023 |
|  | Condition | 5.95 | 2 | .051 | .027 | **Condition** | 10.14 | 2 | .006 | .055 |
| N2 | **Group** | 15.69 | 1 | < .001 | .094 | **Condition** | 47.85 | 2 | < .001 | .082 |
|  | **Age band** | 13.95 | 1 | < .001 | .110 |  |  |  |  |  |
|  | **Condition** | 32.16 | 2 | < .001 | .082 |  |  |  |  |  |
|  | **Group × Condition** | 6.45 | 1 | .011 | .000 |  |  |  |  |  |

*Note.* The best fitting models for LAER component amplitude and latency were all an acceptable fit (Amplitude: P1 [AIC = 855.18, $R^2_c$=0.527], N1 [AIC = 496.71, $R^2_c$=0.439], P2 [AIC = 431.23, $R^2_c$=0.627], N2 [AIC = 1072.4, $R^2_c$=0.603]; Latency: P1 [AIC = 1967.5, $R^2_c$=0.271], N1 [AIC = 958.23, $R^2_c$=0.230, P2 [AIC = 797.32, $R^2_c$=0.886] and N2 [AIC = 2176.2, $R^2_c$=0.45]). Effects that were significant are shown in boldface.

DOI: https://doi.org/10.7554/eLife.46965.024

For P1 amplitude, the best fitting model included the main effect of Condition only. P1 was significantly larger in the Speech than both the Nonspeech and Speech-like conditions, *p*s <0.01, which, in turn, did not differ from each other, p=0.993. For P1 latency, the final model included the main effect of Group in addition to Condition. P1 was significantly later (on average by 11 ms) in the MM group relative to controls, p=0.012, and significantly later in the Speech than both the Nonspeech and Speech-like conditions, *p*s <0.05, which again did not differ, p=0.232. No other main effects were significant for P1.

For N1, the best fitting model for amplitude included the main effects of Age band, Condition and their two-way interaction. The effect for N1 to be larger in the Y age band compared to the O age band just missed significance in the Speech condition (p=0.056). There was no significant difference in Age band for either the Speech nor Speech-like conditions (both *p*s >0.10). For N1 latency, the best fitting model included the main effects of Age band and Condition. Older children showed an earlier N1 than younger children (p=0.005). Moreover, N1 was significantly earlier in the Nonspeech condition than both Speech-like and Speech conditions (both *p*s <0.001), which did not differ from each other (p=0.544).

For P2 amplitude, the best fitting model included the main effects of Age band and Condition, the latter just missing significance (p=0.051) but suggesting a larger response in Nonspeech than in the Speech-like condition (p=0.022). Older children exhibited a larger P2 than younger children (p=0.008). For P2 latency, the final model included the main effects of Group (which just missed significance) and Condition. P2 was earlier (by 10.9 ms) in controls than in children with MMHL (p=0.051). In addition, P2 was earlier in the Nonspeech condition than in both the Speech-like (p=0.002), and Speech (p=0.027) conditions, the latter of which did not differ from each other (p=0.656).

For N2 amplitude, the best fitting model included the main effects of Group, Age band, and Condition, as well as the Group × Condition interaction. Overall, the N2 was larger (by 2.44 µV) in younger than older children (p<0.001). The main effects of Group and Condition have to be considered in light with their two-way interactions. Control children had a

significantly larger N2 than children with MMHL, an effect that was greater in the Speech-like (p<0.001) than both Nonspeech and Speech conditions (both *ps* < 0.05). The best fitting model for N2 latency included a main effect of Condition only. N2 was significantly earlier in the Nonspeech relative to both Speech-like and Speech conditions (*ps* < 0.001; Speech vs. Speech-like: p=0.663).

## Mismatch Responses

A series of logistic regressions were computed on MMN/LDN presence, with Group, Age band and Condition, as well as their two- and three-way interactions as predictors (see ***Appendix 1—table 1***). For the MMN, the best fitting model included Group as a main effect [$\chi^2(1)$=7.39, p=0.007, AIC = 369.4, $R^2_c$=0.033]. Children in the MM group were less likely to show an MMN than children in the NH group, OR = 0.51, *p* = .007, although around half the children with MMHL showed a significant MMN. For the LDN, the best fitting model [$\chi^2(1)$ =1.33, p=0.247, AIC = 243.2, $R^2_c$=0.009] contained no significant main effects or interactions.

Linear mixed-effects models (lme4 package of R; ***Bates et al., 2014***) were then conducted to determine whether Group, Age band, or Condition predicted MMN/LDN amplitude for the large deviants (see ***Appendix 1—table 3***). The best fitting model for MMN amplitude [AIC = 1418.1, $R^2_c$=0.27] included the main effects of Group [$\chi^2(1)$=14.48, p<0.001, $R^2_m$ = .080], Age band [$\chi^2(1)$=5.31, p=0.021, $R^2_m$=0.036], and Condition [$\chi^2(1)$=7.73, p=0.021, $R^2_m$ = .033], as well as a Group × Age band interaction that just missed significance [$\chi^2(1)$ =3.29, p=0.069, $R^2_m$ = .013]. Again, this was driven by a small and non-significant group difference in MMN amplitude between the NH-Y and the MM-Y subgroups, t (135.45) = −1.82, p=0.071, CI [−2.27, 0.09], along with a significant group difference between the NH-O and MM-O subgroups, t(120.56) = −5.15, p<0.001, CI [−3.84,–1.71], driven by a smaller MMN in the MM-O subgroup relative to their NH peers. The best fitting model for LDN amplitude included none of the main effects nor interactions [AIC = 933.4, $R^2_c$=0.010].

We then repeated these analyses only for those individuals who showed a significant MMN or LDN. For MMN amplitude, the best fitting model included Group, Age band, Condition, and Group ×Age band interaction [AIC = 679.42, $R^2_c$ = 0.244]. The main effect of Age band (p=0.020) has to be considered in light with the marginally significant Group ×Age band interaction [$\chi^2(1)$=3.67, p=0.055, $R^2_m$ = .021]. Post-hoc *t*-tests indicated that there was no significant Group difference in the younger children [t(69.57)=−0.14, p=0.891, CI = −1.29, 1.13]. However, older children with MMHL had a significantly smaller MMN than older NH children [t(48.46)=−3.20, p=0.002, CI = −3.01,–0.69]. Last, there was a significant main effect of Condition [$\chi^2(2)$=10.52, p=0.005, $R^2_m$ = .081], showing that MMN amplitude was larger in the Nonspeech and Speech-like than Speech condition (both *ps* <0.05). There was no significant difference between the Nonspeech and Speech-like conditions (p=0.456). The best fitting model for LDN amplitude included none of the main effects nor interactions as predictors [AIC = 345.4, $R^2_c$=0.00].

## Relations with Severity of SNHL

In order to determine whether severity of SNHL predicted presence/absence of components, a series of mixed-effects logistic regression models was applied to determine whether BEPTA threshold, Age band, Condition or Component predicted the presence of a LAER peak in children with MMHL (see ***Appendix 1—table 5***). The same fitting strategy was used as reported earlier. The best fitting model [$\chi^2(7)$=173.67, *AIC* = 600.12, $R^2_c$ = 0.346] included significant main effects of BEPTA [$\chi^2(1)$=57.43, p<0.001, $R^2_m$=0.132], Component [$\chi^2(3)$ =118.17, p<0.001, $R^2_m$=0.240], and their two-way interaction [$\chi^2(3)$=9.30, p=0.025, $R^2_m$=0.001]. The main effects of BEPTA and ERP Component have to be considered in light with their interaction, which indicated that higher BEPTA thresholds had a larger effect on both the P1 and N2 (both *ps* <$1e^{-9}$) than on N1 and P2 components (both *ps* <$10e^{-6}$).

**Appendix 1—table 5.** Results of the logistic regression analyses (MM group only) for the effect of BEPTA on presence of LAER components and MMRs (cross-sectional study).

| Component | Effects | $\chi^2$ | Df | p | $R^2_m$ |
|---|---|---|---|---|---|
| LAER | Component | 118.17 | 3 | < .001 | .240 |
| | BEPTA | 57.43 | 1 | < .001 | .132 |
| | BEPTA × Component | 9.30 | 3 | .025 | .001 |
| MMN | BEPTA | 5.36 | 1 | .020 | .048 |
| LDN | BEPTA | 0.34 | 1 | .559 | .070 |
| | Age band | 0.23 | 1 | .633 | .069 |
| | Age band × BEPTA | 4.23 | 1 | .039 | .065 |

*Note.* The best fitting models for LAER, MMN and LDN amplitude were all an acceptable fit (LAER [AIC = 600.12; $R^2_c$=0.346]; MMN [AIC = 188.9, $R^2_c$=0.05]; LDN [AIC = 122.1, $R^2_c$=0.073]).

DOI: https://doi.org/10.7554/eLife.46965.025

A series of mixed-effects linear regression models was then applied to determine whether BEPTA, Age band or Condition predicted the amplitude and latency of the LAER components, where present, in children with MMHL (see *Appendix 1—table 6*). For P1 amplitude, the best fitting model included BEPTA, Condition and their two-way interaction. Higher BEPTA thresholds were associated with smaller P1 amplitudes in the Speech-like (p=0.037) but not in the Nonspeech nor Speech conditions (both *ps* >0.10). For P1 latency, the best fitting model included BEPTA and Age band as main effects, as well as their two-way interaction. However, neither of the main effects (both *ps* >0.50) nor the interaction (p=0.086) reached significance. For N1 amplitude, none of the main effects or interaction were significant predictors of the best fitting model (p=0.572). For N1 latency, the best fitting model included BEPTA and Condition. Children with higher BEPTA thresholds showed later N1s (p=0.049). For P2 amplitude and latency, none of the main effects or interaction were significant predictors of the best fitting model (*ps* >0.05). For N2 amplitude, the best fitting model included both Age band and Condition as main effects. Younger children showed a larger N2 than older children (p=0.028). Amplitude of the N2 was larger in the Speech than both Speech-like and Nonspeech (both *ps* <0.05), which did not significantly differ from each other (p=0.564). For N2 latency, the best fitting model included only Condition as a main effect. Responses were significantly earlier in the Nonspeech than in both the Speech-like and Speech conditions (both *ps* <0.01), which did not differ from each other (p>0.50).

**Appendix 1—table 6.** Results of the linear mixed-effect analyses (MM group only) for the amplitude and latency of each LAER component (P1, N1, P2, N2) where present, and for MMN and LDN amplitude for all children (cross-sectional study).

| Component | Amplitude | | | | | Latency | | | | |
|---|---|---|---|---|---|---|---|---|---|---|
| | Effects | $\chi^2$ | Df | p | $R^2_m$ | Effects | $\chi^2$ | Df | p | $R^2_m$ |
| P1 | BEPTA | 0.76 | 1 | .380 | .055 | BEPTA | .372 | 1 | .541 | .039 |
| | Condition | 7.89 | 2 | .019 | .101 | Age band | .061 | 1 | .804 | .035 |
| | BEPTA × Condition | 6.48 | 2 | .039 | .044 | BEPTA × Age band | 2.94 | 1 | .086 | .034 |
| N1 | BEPTA | .32 | 1 | .572 | .009 | BEPTA | 6.00 | 2 | .049 | .126 |
| | | | | | | Condition | 3.91 | 1 | .047 | .080 |
| P2 | BEPTA | 2.14 | 1 | .153 | .091 | Condition | 4.72 | 2 | .094 | .037 |
| N2 | Age band | 9.16 | 2 | .010 | .064 | Condition | 8.73 | 2 | .013 | .071 |

*Appendix 1—table 6 continued on next page*

**Appendix 1—table 6 continued**

| | | Amplitude | | | | Latency |
|---|---|---|---|---|---|---|
| | Condition | 4.80 | 1 | .028 | .069 | |
| MMN | Age Band | 10.48 | 1 | .001 | .150 | |
| | BEPTA | 8.45 | 1 | .003 | .115 | |
| LDN | Condition | 2.24 | 2 | .326 | .004 | |

*Note.* The best fitting models for LAER component amplitude and latency, and MMN amplitude were all an acceptable fit (Amplitude: P1 [AIC = 377.99, $R^2_c$ = .45], N1 [AIC = 185.16, $R^2_c$ = .24], P2 [AIC = 142.45, $R^2_c$ = .75], N2 [AIC = 480.39, $R^2_c$ = .43], MMN [AIC = 304.35, $R^2_c$ = .302] and LDN [AIC = 167.35, $R^2_c$ = .005]; Latency: P1 [AIC = 876.47, $R^2_c$ = .17], N1 [AIC = 357.94, $R^2_c$ = .23], P2 [AIC = 268.42, $R^2_c$ = .91], and N2 [AIC = 483.19, $R^2_c$ = .40]). Effects that were significant are shown in boldface

DOI: https://doi.org/10.7554/eLife.46965.026

# Longitudinal study (Time 2)

## Modelling Results

### Late Auditory Evoked Responses

A series of mixed-effects logistic regressions were conducted to determine whether Age band (MM-Y vs. MM-YO), Condition, or Component predicted the presence/absence of a LAER peak. The best fitting model [$\chi^2$(8)=75.15, p<0.001, *AIC* = 366.66, $R^2_c$=0.28] included the main effects of Age band [$\chi^2$(1)=5.53, p=0.018, $R^2_m$=0.10], Component [$\chi^2$(3)=47.07, p<0.001, $R^2_m$=0.27], and an Age band x Component interaction [$\chi^2$(3)=20.84, p<0.001, $R^2_m$=0.09]. The two main effects need to be considered in light of their interaction, which was driven by a decrease in the number of children in the MM-Y subgroup showing a P1 and an N2 as they grew older (MM-YO subgroup) (OR = 0.52, $\chi^2$(1)=5.37, p=0.021, and OR = 0.43, $\chi^2$(1)=5.49, p=0.019, respectively), and an increase in the number of these children showing a P2, (OR = 4.32, $\chi^2$(1)=6.35, p=0.011). The increase in the proportion of children showing an N1 between time-points just failed to reach significance [OR: 2.75, $\chi^2$(1) =3.40, p=0.062].

### Mismatch Responses

The effect of development on presence/amplitude of the MMN was assessed in four ways. First, in order to determine whether children with MMHL were less likely to show a significant MMN as they grew older, a within-subject logistic regression, with Age band (MM-Y vs. MM-YO), Condition, and their two-way interaction as predictors was performed (see **Appendix 1—table 8**). The best fitting model included only Age band as a main effect [$\chi^2$(1) =4.29, p=0.039, AIC = 107.1, $R^2_c$ = .061], thus confirming our hypothesis that children in the MM-YO group were less likely to show an MMN than their younger selves (MM-Y), OR = 0.38, p=0.039.

Second, to determine whether the amplitude of the MMN decreased with age in children with MMHL, a series of within-subjects linear mixed models was run with Age band (MM-Y vs. MM-YO), Condition, and their interaction as predictors for MMN amplitude (see **Appendix 1—table 8**). The final model contained a significant main effect of Age band only [$\chi^2$(1)=4.80, $R^2_c$ = .000]. MMN amplitude was significantly smaller in the MM-YO subgroup than in the MM-Y subgroup (p=0.028).

Third, to assess whether children in the MM-YO subgroup were less likely to show an MMN compared to those from Time 1, a between-subject logistic regression was performed across the three older groups of children (MM-YO, MM-O, NH-O) as a function of Condition (see **Appendix 1—table 8**). The best fitting model included only Group as a significant predictor [$\chi^2$(2)=16.98, p<0.001, *AIC* = 212.9, $R^2_c$=0.112]. Children in the NH-O were more

likely (OR = 5.71, $\chi^2$(1)=15.65, p<0.001) to show an MMN than children in the MM-YO group, whereas children in the MM-YO and MM-O subgroups did not differ (OR = 1.83, $\chi^2$(1)=0.22, p=0.141).

Finally, to assess whether children in the MM-YO subgroup showed an MMN that was reduced in amplitude relative to the two other older groups of children, a series of linear mixed-effect analyses was run comparing the amplitude of the MMN evoked in the MM-YO, MM-O, and NH-O subgroups as a function of Condition (see **Appendix 1—table 8**). The best fitting model contained the two main effects of Subgroup [$\chi^2$(2)=23.25, p<0.001, $R^2_m$=0.168] and Condition [$\chi^2$(3)=10.10, p=0.018, $R^2_m$=0.041]. On average, the MM-YO subgroup showed an MMN that was significantly smaller than that of the NH-O subgroup (p<0.001), but no different from that of the initial MM-O subgroup (p=0.957). The MMN was larger in both Nonspeech and Speech-like than in the Speech condition (both *ps* <0.05), but the Nonspeech and Speech-like conditions did not differ from each other (p>0.50).

**Appendix 1—table 7.** Periods of the 100–600 ms post-stimulus epoch (in ms) that contained a significant (p<0.01) MMN (100–400 ms) or LDN (400–600 ms) for each Condition (nonspeech, speech-like, speech), Deviant type (small vs. large), Group (CA vs. MM), and Age band (Y vs. O) (longitudinal study).

|  |  | Younger (Y) |  | Older (O) |  |
| --- | --- | --- | --- | --- | --- |
| Condition | Deviant | MM-Y (n = 13) | MM-YO (n = 13) | MM-O (n = 23) | NH-O (n = 18) |
| Nonspeech | Small | - | - | - | - |
|  | Large | 162–260 378–400 566–600 | - | - | 120–220 288–370 |
| Speech-like | Small | - | - | - | - |
|  | Large | - | - | - | 162–280 424–494 |
| Speech | Small | - | - | - | - |
|  | Large | 310–370 | 296–368 412–476 | 468–576 | 236–326 |

*Note*. Differences in the values obtained between for the MM-Y groups in **Table 2** and here are due to slightly different pre-processing strategies, together with a change in MM-Y group sample between Time 1 (n = 23) and Time 2 (n = 13).

DOI: https://doi.org/10.7554/eLife.46965.027

**Appendix 1—table 8.** Results of the logistic and linear regression analyses for MMN presence/absence and amplitude (longitudinal study).

|  | Logistic regression |  |  |  |  | Linear regression |  |  |  |  |
| --- | --- | --- | --- | --- | --- | --- | --- | --- | --- | --- |
| Type | Effects | $\chi^2$ | Df | p | $R^2_m$ | Effects | $\chi^2$ | Df | p | $R^2_m$ |
| Within-subject | **Group** | **4.24** | **76** | **.039** | **.065** | **Group** | **4.80** | **1** | **.028** | **.060** |
| Between-subject | **Group** | **17.0** | **159** | **< .001** | **.124** | **Group** | **23.2** | **2** | **< .001** | **.168** |
|  |  |  |  |  |  | **Condition** | **10.01** | **3** | **.018** | **.041** |

*Note*: Within-subject analyses included MM-YO and MM-Y as groups. Between-subject analyses included MM-YO, MM-O and NH-O. The best fitting models for MMN presence/absence and amplitude were all an acceptable fit (Presence: Within-subject [AIC = 107.1, $R^2_c$ = .06], Between-subject [AIC = 212.98, $R^2_c$ = .12]; Amplitude: Within-Subject [AIC = 398.2, $R^2_c$ = .06], Between-subject [AIC = 791.0, $R^2_c$ = .315]). Effects that were significant are shown in boldface.

DOI: https://doi.org/10.7554/eLife.46965.028

**Appendix 1—table 9.** Latencies (ms) of the MMN/LDN evoked by large deviants for the NH group for each age band and condition (cross-sectional and longitudinal studies).

| Study | Condition | MMN | | LDN | |
|---|---|---|---|---|---|
| | | Y | O | Y | O |
| Cross-sectional | Nonspeech | 164 | 160 | | |
| | Speech-like | 264 | 232 | 430 | 444 |
| | Speech | 284 | 262 | 486 | 510 |
| Longitudinal | Nonspeech | - | 168 | | |
| | Speech-like | - | 230 | - | 484 |
| | Speech | - | 266 | - | 508 |

DOI: https://doi.org/10.7554/eLife.46965.029

