## [Decision Letter]

Thank you for submitting your article "Functional Brain Alterations Following Mild-to-Moderate Sensorineural Hearing Loss in Children" for consideration by *eLife*. Your article has been reviewed by three peer reviewers, and the evaluation has been overseen by a Reviewing Editor and Barbara Shinn-Cunningham as the Senior Editor. The following individuals involved in review of your submission have agreed to reveal their identity: Dan H Sanes (Reviewer #2); David Moore (Reviewer #3).

The reviewers have discussed the reviews with one another and the Reviewing Editor has drafted this decision to help you prepare a revised submission.

Summary:

The purpose of this project was to investigate the neural processing of sounds across development in children with mild-to-moderate sensorineural hearing loss using EEG. This is particularly important because much of our knowledge to-date is based on either more severe forms of hearing deprivation and/or cross-sectional studies, so examining auditory responses longitudinally in children with mild-to-moderate hearing loss is an important feature. The results suggest children with mild-to-moderate hearing loss show changes in amplitude and latency in MMN components.

Essential revisions:

1) The potential influence of audiometric thresholds on the measures was not adequately addressed (this covers both the stimulus level and degree of hearing loss). Stimuli were delivered at the same level to all children, regardless of hearing loss, and participants who had hearing aids did not wear them. It is true that (putting aside the hearing aid issue for a moment) a constant level to all listeners reflects "real world" listening conditions. At the same time, it substantially weakens the claim about "brain alterations" if the brains are getting stimulated differently. That is, would the brains of older MMHL children produce control-like amplitudes if stimulated at a sufficient SPL? Or, perhaps easier to look at, would listeners with normal hearing produce smaller amplitudes if stimulated at a lower SPL? The critical question is whether the results represent a shift due to simply being at a different place along a stimulation curve, or a fundamental change in brain processing due to hearing impairment.

2) Age is an important factor that was not adequately addressed. For example, if there were hearing changes with age (as might be expected in children with early onset mild-moderate hearing loss) these hearing changes could explain your findings. A clearer examining of how BEPTA shifts with age would be useful, but more helpful would be (a) plotting audiograms for each group and (b) considering information not contained in the BEPTA. Relatedly, what looks like rather parallel performance for the MMHL and NH groups in Figure 3 seems to contradict the idea that effects might be age-dependent. These are some specific examples of a broader need for more clarity on age effects and how these can be accounted for in the results and interpretation.

3) The role of hearing aids needs to be better addressed in interpretation of the results, above the potential contribution to presentation level (point 1). For example, what was the duration of hearing aid use? And how consistently did they use their hearing aids? There is evidence that adolescents tend not to use their hearing aids (e.g., Scherer, 1996), while in children up to age 7, increasing age predicts more consistent hearing aid use (Walker et al., 2013). It is therefore possible that the consistency of treatment that is being received for their hearing loss is also a defining difference between the two age bands in both the cross-sectional and the longitudinal components of this study.

4) In some instances, the descriptions of results in the text did not clearly line up with the data displayed in the figures. For example, a sentence at the end of the Introduction "Despite showing age-appropriate MMNs at the time of Experiment 1, these children showed MMNs that were reduced or absent by Experiment 2." is not in accord with a sentence in the description of the results of Experiment 2 "In the Speech condition, a significant MMN remained in the MM-YO group." Perhaps some of the confusion comes from referring to different conditions in different contexts; however, there was an overall sense of a lack of clarity in the presentation of the results (along with sympathy that there are a lot of variables to look at here). Please clarify the results presentation and make sure the text descriptions clearly line up with results figures as presented.

5) Duration of the analysis window. The analyses focus on the waveforms within the first 400 ms after stimulus onset, but there also appear to be potentially interesting group differences between the waveforms 400-600 ms post-stimulus-onset, particularly for the speech stimuli. Clearer justification for this choice (or better still, an analysis of other time windows) would be useful here.

6) Data and stimuli availability. Even in the absence of ethics approval for sharing personally identifiable information, summary measures should be shareable, as should stimuli used in the experiment. For example, age group, hearing status, and BEPTA figure prominently in the analyses. Similarly the latency and amplitude measures extracted from the raw data don't pose any risk of personal identification. Please see the policies section of the *eLife* Author Guide.

---

## [Author Response]

Essential revisions:1) The potential influence of audiometric thresholds on the measures was not adequately addressed (this covers both the stimulus level and degree of hearing loss). Stimuli were delivered at the same level to all children, regardless of hearing loss, and participants who had hearing aids did not wear them. It is true that (putting aside the hearing aid issue for a moment) a constant level to all listeners reflects "real world" listening conditions. At the same time, it substantially weakens the claim about "brain alterations" if the brains are getting stimulated differently. That is, would the brains of older MMHL children produce control-like amplitudes if stimulated at a sufficient SPL? Or, perhaps easier to look at, would listeners with normal hearing produce smaller amplitudes if stimulated at a lower SPL? The critical question is whether the results represent a shift due to simply being at a different place along a stimulation curve, or a fundamental change in brain processing due to hearing impairment.

This is an important question but unfortunately one that is very challenging to address. Sensorineural hearing loss (SNHL) affects not only the absolute hearing thresholds of affected individuals, but also their dynamic range. That is, many individuals with SNHL experience loudness recruitment for high intensity stimuli, meaning that listeners may not tolerate stimuli at intensities that are matched for sensation level (SL) with those of normally hearing (NH) listeners. Moreover, since the participants tested here were children, we were particularly keen to avoid causing further damage to the residual hearing they had. The result of this is that, in practice, it is very difficult to match the SL of stimuli experienced by children with MMHL to that experienced by NH controls.

One potential way round this would have been to present NH children with lower-intensity stimuli that were matched for SL to those experienced by the MMHL group. However, this again is not straightforward, for three reasons. First, because BEPTA thresholds for children with MMHL range from 21-70 dB HL, there was no consistent SL for participants in the MMHL group. Second, because audiometric thresholds varied between ears for some children with MMHL, there was no consistent between-ear SL for participants in the MMHL group. Third, because audiometric thresholds varied across frequencies for participants in the MMHL, there was no consistent across-frequency SL for this group. The consequence of this was that in order to control for SL, we would have needed to match each individual child with MMHL to a NH child of the same age, and present the latter with stimuli that were shaped in intensity across frequencies and ears. It remains our view that this would have introduced too many moving parts to what was already a complex study.

That said, in a more recent (as yet unpublished) study we have examined the effects of amplification on the MMN in children with MMHL. In this study, 18, 8-to-16 year-old children with MMHL, and 15 NH age-matched controls were tested. Electrophysiological measures were recorded in response to the same standard (probability: 90%) and large (but not small) deviant speech stimuli that were used in the current study. There were however a few methodological differences to the present study, the main one being that for children with MMHL, stimuli were presented both unamplified (70 dB SPL) and with a simulated, personalised amplification (NAL-R formula, limited at 85 dB SPL). NH children were only presented with the unamplified condition. Importantly, preliminary results from this study provide important insights regarding the interpretation of the study reported here. Author response image 1 represents the differential wave (i.e., standards minus deviants) for the MMN observed in NH children presented with unamplified stimuli (left panel), children with MMHL presented with unamplified stimuli (middle panel), and children with MMHL presented with amplified stimuli (right panel). Whereas an MMN was found to be present in a large portion of the response for NH controls, this was not the case for children with MMHL, regardless of whether the stimuli were presented unamplified or amplified. The lack of an MMN in the amplified condition strengthens our claim that MMHL leads to fundamental changes in neural processing and that these findings are not simply the result of differences in the amount of stimulation being received.

2) Age is an important factor that was not adequately addressed. For example, if there were hearing changes with age (as might be expected in children with early onset mild-moderate hearing loss) these hearing changes could explain your findings. A clearer examining of how BEPTA shifts with age would be useful, but more helpful would be (a) plotting audiograms for each group and (b) considering information not contained in the BEPTA. Relatedly, what looks like rather parallel performance for the MMHL and NH groups in Figure 3 seems to contradict the idea that effects might be age-dependent. These are some specific examples of a broader need for more clarity on age effects and how these can be accounted for in the results and interpretation.

We have made several changes to the manuscript which confirm that, for both studies, audiometric thresholds did not deteriorate with age for the MMHL group. Regarding the cross-sectional study, Figure 1 now shows audiometric thresholds (0.25-8 kHz for left and right ears) for children with MMHL compared to NH controls, plotted separately for the younger and older age bands. This illustrates that there were no differences in audiometric thresholds between younger and older children, whatever their group. We now also include statistics to that effect (see Table 1 and subsection “Cross-sectional study (Time 1): Participants”, last paragraph) which report that the MM-O subgroup did not differ from the MM-Y subgroup in BEPTA thresholds, worse ear (WEPTA) thresholds, or mean PTA (MePTA) thresholds. Regarding the longitudinal study, audiometric thresholds for the MM-YO group at Time 1 and Time 2 are now reported in Figure 10—figure supplement 1 and Table 2, and statistical analyses in the last paragraph of the subsection “Longitudinal study (Time 2): Participants”. The MM-YO group did not differ in their BEPTA, WEPTA, or MePTA thresholds between Time 1 and Time 2.

Upon performing these analyses, we also decided to see if the groups also differed on any other audiological measures that we had taken. For the cross-sectional study, the MM-Y and MM-O subgroups did not differ in their age of detection of SNHL, nor in the age at which they were first fitted with hearing aids (see Table 1 and the aforementioned paragraph). However, children in the MM-O subgroup were less likely to have or wear hearing aids (78%) than children in the MM-Y subgroup (100%). Note though that this difference does not apply in the case of the longitudinal study where children in the MM-YO group (all of whom wore hearing aids) were compared to themselves.

Regarding what was Figure 3 in the original submission (now Figure 4—figure supplement 3), as we report in the manuscript, we did not find an interaction between group and age on the latency or amplitude of any of the LAER components (i.e., on P1, N1, P2, N2), where present (see “LAERs and MMNs, but not LDNs, are smaller or later for children with MMHL, when present” and Appendix—table 4). Nor did we see an interaction between group and age regarding the presence/absence of any of the LAER components (see “Children with MMHL show reduced presence of LAER components to sounds”). Rather, our findings suggest that any group × age interaction may be specific to the MMN; namely, that the MMN is (a) less likely to be present at the group level, and (b) where present, is likely to be smaller, in older children with MMHL compared to age-matched NH controls (see Figure 5). Efforts have been taken to improve the clarity of the manuscript for this revision, particularly with regard to the Results section.

3) The role of hearing aids needs to be better addressed in interpretation of the results, above the potential contribution to presentation level (point 1). For example, what was the duration of hearing aid use? And how consistently did they use their hearing aids? There is evidence that adolescents tend not to use their hearing aids (e.g., Scherer, 1996), while in children up to age 7, increasing age predicts more consistent hearing aid use (Walker et al., 2013). It is therefore possible that the consistency of treatment that is being received for their hearing loss is also a defining difference between the two age bands in both the cross-sectional and the longitudinal components of this study.

As outlined above, Table 1 now presents the mean age of detection of MMHL, the number of children who were fitted with hearing aids, and the mean age of aiding for those children with MMHL who wore hearing aids, for the Y and O subgroups. Regarding the consistency of hearing aid use in the MM groups, unfortunately we do not have much data on this as we were not able to access hearing aid logs, and parent report measures of hearing aid use have been shown to be unreliable (Munoz, Preston, and Hicken, 2014, Journal of the American Academy of Audiology 25(4):380-7). Nonetheless, there was some evidence for lower uptake of hearing aids in the MM-O subgroup at Time 1, in that three children in this group had not been prescribed with hearing aids, and two were refusing to wear their hearing aids. However, this may reflect a historical point in time; children in this subgroup were born before the UK national roll-out of the Universal Newborn Hearing Screening Programme. All of the children in the MM-Y subgroup at Time 1 had been prescribed with hearing aids, and all were wearing them at Time 2, although again, we do not know how consistently. We now include some discussion about this in the Discussion section (subsection “Functional changes following mild to moderate hearing loss”, last paragraph).

4) In some instances, the descriptions of results in the text did not clearly line up with the data displayed in the figures. For example, a sentence at the end of the Introduction "Despite showing age-appropriate MMNs at the time of Experiment 1, these children showed MMNs that were reduced or absent by Experiment 2." is not in accord with a sentence in the description of the results of Experiment 2 "In the Speech condition, a significant MMN remained in the MM-YO group." Perhaps some of the confusion comes from referring to different conditions in different contexts; however, there was an overall sense of a lack of clarity in the presentation of the results (along with sympathy that there are a lot of variables to look at here). Please clarify the results presentation and make sure the text descriptions clearly line up with results figures as presented.

The Results section has now been substantially revised with a view to improving clarity. We have endeavoured to ensure that the main results of the study are conveyed clearly and that overall findings are not over-simplified .

5) Duration of the analysis window. The analyses focus on the waveforms within the first 400 ms after stimulus onset, but there also appear to be potentially interesting group differences between the waveforms 400-600 ms post-stimulus-onset, particularly for the speech stimuli. Clearer justification for this choice (or better still, an analysis of other time windows) would be useful here.

This is a good suggestion and something we had not noticed before as we had been focusing on the conventional time window for the MMN (i.e., 100-400 ms post-stimulus onset). Responses that occur after this time window are more typically considered to be akin to the late discriminatory negativity (LDN). We now include analyses of this response in the paper and associated discussions therein.

6) Data and stimuli availability. Even in the absence of ethics approval for sharing personally identifiable information, summary measures should be shareable, as should stimuli used in the experiment. For example, age group, hearing status, and BEPTA figure prominently in the analyses. Similarly the latency and amplitude measures extracted from the raw data don't pose any risk of personal identification. Please see the policies section of the eLife Author Guide.

We now provide the non-identifiable data, stimuli, and analysis scripts (see https://github.com/acalcus/MMHL.git).